

# TEMIS UV product validation using NILU-UV ground-based measurements in Thessaloniki, Greece

Melina-Maria Zempila[1*,2], Jos H. G. M. van Geffen[3], Michael Taylor[2], Ilias Fountoulakis[2],
Maria-Elissavet Koukouli[2], Michiel van Weele[3], Ronald J. van der A[3], Alkiviadis Bais[2],
Charikleia Meleti[2], and Dimitrios Balis[2]

[1]USDA UV-B Monitoring and Research Program, Colorado State University, Fort Collins, Colorado 80523, U.S.A.
[2]Laboratory of Atmospheric Physics, Aristotle University of Thessaloniki, PO Box 149, 54124, Thessaloniki, Greece.
[3]Royal Netherlands Meteorological Institute (KNMI), De Bilt, The Netherlands.

*Correspondence to:* Melina Maria Zempila (melina.zempila@colostate.edu)

**Abstract.** This study aims to cross-validate ground-based and satellite-based models of three photobiological UV effective dose products: the CIE erythemal UV, the production of vitamin D in the skin and DNA-damage, using high temporal resolution surface-based measurements of solar UV spectral irradiances from a synergy of instruments and models. The satellite-based Tropospheric Emission Monitoring Internet Service (TEMIS; version 1.4) UV daily dose data products were evaluated over

the period 2009 to 2014 with ground-based data from a NILU-UV multifilter radiometer located at the Northern mid-latitude super site of the Laboratory of Atmospheric Physics, Aristotle University of Thessaloniki (LAP/AUTh) in Greece.

For the NILU-UV effective dose rates retrieval algorithm, a neural network (NN) was trained to learn the nonlinear functional relation between NILU-UV irradiances and collocated Brewer-based photobiological effective dose products. Then the algorithm was subjected to sensitivity analysis and validation. The correlation of the NN estimates with target outputs was high

(r=0.988 to 0.990) and with a very low bias (0.000 to 0.011 in absolute units) proving the robustness of the NN algorithm. For further evaluation of the NILU NN derived products, retrievals of the vitamin D and DNA-damage effective doses from a collocated YES UVB-1 pyranometer where used. For cloud free days, differences in the derived UV doses are better than 2% for all UV doses products, revealing the reference quality of the ground-based UV doses at Thessaloniki from the NILU-UV NN retrievals.

The TEMIS UV doses used in this study, are derived from ozone measurements by the SCIAMACHY/Envisat and GOME2/MetOp-A satellite instruments, over the European domain in combination with SEVIRI/Meteosat based diurnal cycle of the cloud cover fraction per $0.5° \times 0.5°$ ($lat \times lon$) grid cells. TEMIS UV doses were found to be ∼12.5% higher than the NILU NN estimates but, despite the presence of a visually apparent seasonal pattern, the $R^2$ values were found to be robustly high and equal to 0.92-0.93 for 1,588 all sky coincidences. These results significantly improve when limiting the dataset to cloud

free days with differences of 0.57% for the erythemal doses, 1.22% for the vitamin D doses and 1.18% for the DNA-damage doses, with standard deviations of the order of 11-13%. The improvement of the comparative statistics under cloud-free cases further testifies to the importance of the appropriate consideration of the contribution of clouds in the UV radiation reaching the Earth's surface. For the urban area of Thessaloniki with highly variable aerosol the weakness of the implicit aerosol information introduced to the TEMIS UV dose algorithm was revealed by comparison of the datasets to aerosol optical depths



at 340 nm as reported by a collocated CIMEL sunphotometer, operating in Thessaloniki at LAP/AUTh as part of the NASA
Aerosol Robotic Network.

# 1   Introduction

During the last few decades, the danger of overexposure to UV sunlight has been well analysed and a causal link has been
established to skin diseases and cancer since the mutation of DNA can be triggered by extreme UV-B doses (Xiang et al., 2014;
Parkin et al., 2011; Berwick et al., 2005; Setlow, 1974, among others). Furthermore, the cutaneous production of vitamin D
is also activated by spectral UV radiation, hence accurate knowledge of 'safe' UV doses for humans is paramount (McKenzie
et al., 2009; Webb et al., 1988; MacLaughlin et al., 1982, among others). Of particular relevance is the Commission Interna-
tionale de l' Éclairage (CIE) action spectrum as a model for the susceptibility of skin to sunburn (erythema) (McKinlay and
Diffey, 1987). As a result of advances in the fields of photobiology and ground-based measurements of UV using different types
of instrumentation, a variety of methods now exist to obtain erythemal, vitamin D and DNA-damage dose rates (Kazantzidis
et al., 2009; Webb and Engelsen, 2006; Pope et al., 2008; Engelsen et al., 2005; Samanek et al., 2006).

In parallel, space technology has been making huge steps forward to monitor the Earth 's surface and atmosphere at higher
spatial and temporal resolution and erythemal, vitamin D and DNA-damage dose rates and doses can now be retrieved globally
from solar backscattered radiation observations from different satellite sensors. Subsequently, long, reliable and high tempo-
ral resolution ground-based estimates of surface photobiological effective dose quantities are of high importance in order to
validate and characterize the satellite-derived UV products. Ozone layer depletion and recovery in times of climate change
reinforce the need for establishing global long-term and quality assured climate data records of the incident solar UV daily
doses at the surface.

In this study, photobiological UV daily doses retrieved from ground-based measurements using empirical models and satellite
estimates are cross-validated to assess their accuracy and potential utility.





## 2 TEMIS satellite-based UV data products

### 2.1 Operational services

The Tropospheric Emission Monitoring Internet Service (TEMIS) was established in 2001 at the Royal Netherlands Meteorological Institute (KNMI) as part of a project from the European Space Agency (ESA) and the service has been maintained since.

The TEMIS UV data product services, started in 2003, are available through the webportal at http://www.temis.nl/uvradiation/. The UV products, currently version 1.4, are produced in near-real time on a latitude × longitude grid of $0.5° \times 0.5°$ and consist of data sets, maps, and time series. The products are calculated using operational satellite data streams of the global ozone distribution and, over Europe, the diurnal variation in cloud cover fraction.

The TEMIS UV data products essentially exploit the empirically-based parametrisation by Allaart et al. (2004) of the amount

of UV radiation incident at the surface in $W/m^2$, as function of the total ozone column and the solar zenith angle at a given local solar time, taking into account an appropriate action spectrum, i.e. the wavelength dependent response to UV radiation of health effects or otherwise.

Since the initiation of the TEMIS UV services maintenance and updates were implemented following changes in e.g. the operational assimilated global ozone distribution (Eskes et al., 2003), based on first the SCIAMACHY instrument aboard ENVISAT

(Bovensmann et al., 1999) up to April 2012, and later GOME-2 aboard MetOp-A (Hassinen et al., 2016). Recently, the global ozone Multi-Sensor Reanalysis version 2 (MSR-2) by van der A et al. (2015) has been used to create a reanalysis of the global clear-sky UV index for a longer historical period (from November 1978 to December 2012).

Cloud attenuation over Europe is prescribed using the near-real time cloud mask product (Derrien and Le Gléau, 2005) provided by the EUMETSAT Nowcasting Satellite Application Facility (NWC-SAF), which is received, processed and archived

at KNMI since July 2005. The operational cloud cover data set has been based on the different SEVIRI instruments operational aboard the Meteosat Second Generation (MSG) satellites from January 2004 onwards using the Meteosat 8, 9 and 10 platforms, respectively. The effect of grid cell average surface elevation, though not the actual 3-D topography, on surface UV is taken into account in the calculations. Changes in surface albedo are prescribed using a monthly climatology of surface reflectivity (Herman and Celarier, 1997). The effects of aerosols are included implicitly in the parameterization but do not vary over time

(Badosa and van Weele, 2002).

### 2.2 Products and algorithms

TEMIS provides two types of surface UV products: (i) the clear-sky erythemal UV index and (ii) the daily UV dose (daily integral) related to different health effects. The erythemal UV index (UVI-CIE) is determined using the action spectrum adopted

by the International Commission on Illumination (CIE) for erythema or reddening of the skin due to sunburn (McKinlay and Diffey, 1987). Following international agreements, the UVI-CIE represents the amount of UV radiation at local solar noon, i.e. when the sun is highest in the sky, under clear-sky conditions. The UVI-CIE is usually given as a dimensionless index, where 1 unit equals $25 \, mW/m^2$. Using the operational meteorological data streams (temperature, pressure, winds) which are



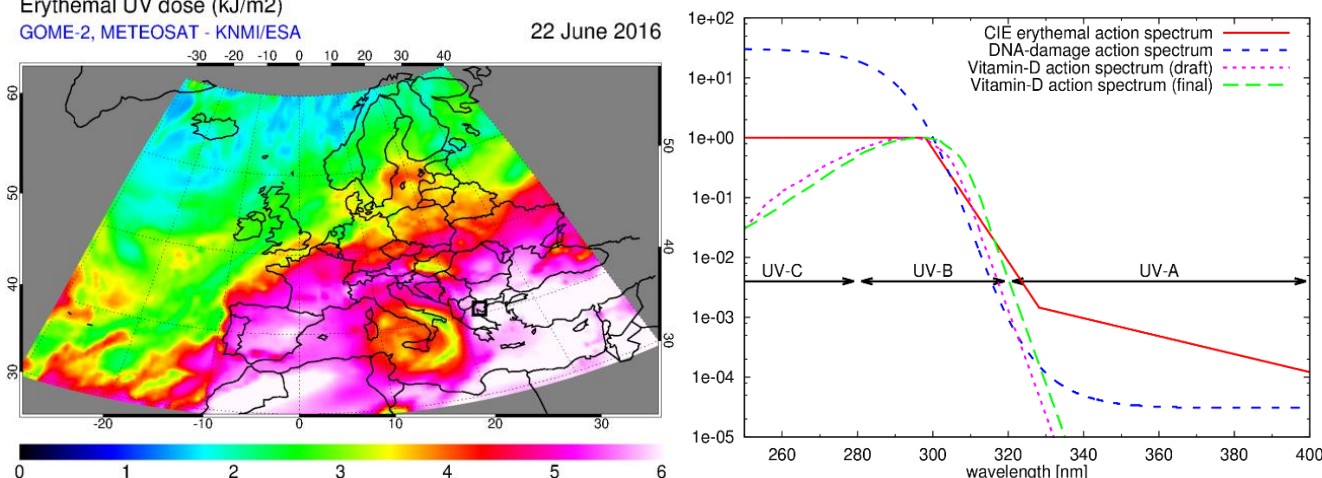

**Figure 1.** Erythemal UV dose over Europe on 22 June 2016. Thessaloniki, indicated by a black square, had an almost cloud-free day with an erythemal UV dose of 5.77 kJ/m$^2$ and an erythemal UV index of 10.1 (left panel). Action spectra of erythema (red solid), generalized DNA-damage (blue short-dashed), and production of vitamin D (magenta dotted: draft version as used within TEMIS (Holick et al., 2005), and green long-dash: final version as adopted by the CIE (Bouillon et al., 2006) (right panel).

included in the ozone data assimilation (Eskes et al., 2003), the UVI-CIE is available in forecast mode and TEMIS provides forecasts of both the global ozone field and UVI-CIE for today and the coming 8 days.

The daily UV dose (UVD) is the total amount of UV radiation, usually given in kJ/m$^2$, integrated between sunrise and
5  sunset, accounting for the variation in the solar zenith angle (SZA) and cloud cover fraction (in TEMIS (version 1.4) this is available over Europe only) during the day, see Figure 1, left. The UV dose is calculated for three action spectra (see Figure 1, right) : the erythemal UV dose (UVD-CIE) based on the CIE erythemal action spectrum (McKinlay and Diffey, 1987), identical to the one used for the UVI-CIE; the generalised DNA-damage UV dose (UVD-DNA) based on the action spectrum determined by Setlow (1974) and normalised at 300 nm based on Bernhard and Seckmeyer (1997); the vitamin D UV dose (UVD-VitD)
10  based on the action spectrum for the production of previtamin-D3 in the human skin (Holick et al., 2005).

Note that the 2005 (draft) version by Holick et al. (2005) used for UVD-VitD within TEMIS differs slightly from the CIE adopted vitamin D action spectrum (Bouillon et al., 2006), see Figure 1, right. The difference, which includes a wavelength shift, would increase the TEMIS data by a factor of about 2.2 (2.1 in summer, 2.3 in winter) when using the CIE vitamin D action spectrum - a change that will be implemented in a forthcoming update of the TEMIS UV operational data streams.

15  For each action spectrum, a parametrisation is applied following Allaart et al. (2004) for the UV solar irradiance as a function of SZA($t$) and total ozone column, providing a 'bare' UV index (UVI$'$) at time $t$ using the global assimilated ozone field at local solar noon ($t = 12$h). The UV index at time $t$ is then found from UVI$'$ after applying a set of correction factors:

$$\mathrm{UVI}(t) = \mathrm{UVI}'(t) \cdot f_D \cdot f_C \cdot f_H \cdot f_A \qquad [\mathrm{W/m}^2] \tag{1}$$



where $f_D$ is the correction for the day-to-day variation in the Sun-Earth distance, $f_C$ the correction for the attenuation due to clouds (in case of clear-sky conditions: $f_C = 1$), $f_H$ the correction for the surface elevation, and $f_A$ the correction for the ground albedo. At this point it should be mentioned that the TEMIS products' uncertainty can currently be estimated only from the errors reported in the ozone total amount, thus it only reflects the lower boundary of the errors seen in the UV doses. Based on this fact, TEMIS products include an uncertainty of 2-3% in the daily doses.

The UV index at local solar noon, $\mathrm{UVI}(t = 12\mathrm{h})$, follows directly from Eq. (1) after division by $25~\mathrm{mW/m^2}$. The UVD products, in $\mathrm{kJ/m^2}$, are determined from a 10-min step integration of $\mathrm{UVI}(t)$ between sunrise and sunset, which are assumed to lie symmetrical around local solar noon. For the calculation of $f_C$ the NWC-SAF cloud mask is converted to a cloud fraction ($C_f$) by counting the clear vs. cloudy instances per UV grid cell of $0.5° \times 0.5°$ (latitude $\times$ longitude). The cloud correction factor in Eq. (1) is then given by:

$$f_C = \begin{cases} 1.0 & , \quad C_f < 0.02 \\ 0.9651 - 0.2555 \cdot C_f & , \quad 0.02 \le C_f \le 0.98 \\ 0.5 & , \quad C_f > 0.98 \end{cases} \tag{2}$$

a relationship that has been determined from the effect of clouds on surface UV at the location of KNMI at De Bilt in The Netherlands (van Geffen et al., 2004; van Weele et al., 2005). For the calculation of $f_H$ a 5% increase of the incident UV irradiance per $\mathrm{km}$ surface elevation above sea level is assumed:

$$f_H = 1 + 0.05 \cdot H \tag{3}$$

where the surface elevation $H$ (in km) is determined from the GTOPO30 database (https://lta.cr.usgs.gov/GTOPO30/), resampled to the $0.5° \times 0.5°$ UV grid. For the calculation of $f_A$ the following function of ground albedo ($A_g$) is applied, taking into account multiple reflections between the surface and the overlying atmosphere:

$$f_A = \frac{1 - 0.25 \cdot 0.09}{1 - 0.25 \cdot A_g} \tag{4}$$

The function derives from the series $1 + xy + (xy)^2 + \cdots = 1/(1 - xy)$ where $x = 0.25$ is the UV albedo of the overlying atmosphere for upward reflected UV radiation and $y = A_g$. Since the Allaart et al. (2004) UV index parametrisation is empirically based on UV data collected at De Bilt and Paramaribo, the $A_g$ at these (urban) sites − with a 12-month average value of $0.09$ − is used as a normalisation factor for the calculation of $f_A$. The data for $A_g$ at each UV grid cell are taken at $335~\mathrm{nm}$ from the monthly TOMS/GOME climatology, which uses the spectral dependency of the GOME database (Koelemeijer et al., 2003) but with a scaling to match the TOMS 340/380 nm database (Herman and Celarier, 1997; Boersma et al., 2004).

Note that there is no explicit correction in Eq. (1) for the variable presence of aerosols in the TEMIS UV data products. However, the Allaart et al. (2004) empirically-based parametrization includes an implicit aerosol correction due to the average aerosol load over these two urban sites: an AOD at $368~\mathrm{nm}$ of $0.3$ and an aerosol single scattering albedo (SSA) of $0.9$ (Badosa and van Weele, 2002). For situations where the real aerosol load is lower (higher) than that assumed load, the UV data products will underestimate (overestimate) the UV index and UV dose. With potential future near-real time availability of aerosol optical





parameters at a global scale, the correction factors derived by Badosa and van Weele (2002) could be applied within future updates of the TEMIS UV services.

## 3 Ground-based data products

### 3.1 Instruments at Thessaloniki

The calculation of the photobiological doses over Thessaloniki (40.63°E, 22.96°N) are based on measurements taken by three different types of instruments in continuous operation at the Laboratory of Atmospheric Physics of the Aristotle University of Thessaloniki (LAP/AUTh: http://lap.physics.auth.gr).

Firstly, a Brewer MKIII spectrophotometer with serial number #086 (B086) is equipped with a double monochromator and measures the UV solar irradiance spectrum (286.5 - 363 nm) with a wavelength step of 0.5 nm within 7 minutes using a

triangular-like slit that has a full width at half maximum (FWHM) of 0.55 nm. The spectra used in this study have recently been subjected to quality control and re-evaluation (Fountoulakis et al., 2016a) after which the remaining 1-sigma uncertainty is estimated to be 5% (Garane et al., 2006) for wavelengths longer than 305 nm and for solar zenith angles (SZA) smaller than 80°. For lower wavelengths and higher SZA the uncertainty is larger as a consequence of the photon noise that dominates due to the low recorded signal (Fountoulakis et al., 2016b). The simpler, single monochromator Brewer with serial number #005

(B005) has been operational in Thessaloniki since 1982 and has been providing continuous, well-calibrated and documented total ozone column measurements (Bais et al., 1985; Meleti et al., 2012; Zerefos, 1984).

Secondly, a Norsk Institutt for Luftforskning (NILU)-UV multi-filter radiometer has been fully operational in Thessaloniki since 2005 and forms part of the UVNET network of NILU-UV radiometers (http://www.uvnet.gr). The NILU-UV with serial number 04103 (NILU103) provides one-minute measurements in 5 UV channels with nominal central wavelength at 302, 312,

320, 340 and 380 nm and a FWHM of 10 nm; while its sixth channel measures the Photosynthetically Active Radiation (PAR), and is used here to determine cloud-free cases based on the cloud detection algorithm proposed by Zempila et al. (2016b). Although the B086 measures the UV spectrum with high spectral resolution, the time frequency of the scans usually varies from 20 to 40 minutes. Nevertheless, Brewer spectrophotometers are a very powerful means for calibrating other UV measuring instruments that provide higher temporal resolution measurements. By calibrating the NILU103 measurements with the B086

coincident irradiances, we estimate that the uncertainties of the NILU103 irradiance measurements used in this study are less than 5.6% (Zempila et al., 2016a).

Thirdly, a Yankee Environmental System (YES) UVB-1 radiometer operating also in Thessaloniki, provides one minute erythemal dose measurements with a spectral response very similar to the erythemal action spectrum (McKinlay and Diffey, 1987). Using model simulations with the libRadtran radiative transfer (Emde et al., 2016) proper weighting factors are calculated with

respect to SZA and the total ozone column (TOC) and are used to transform the UVB-1 measurements into erythemal irradiance (Lantz et al., 1999). A similar transformation is applied for the Vitamin D and DNA-damage weighted irradiances (see section 3.2.3). In addition, the Brewer measurements have been used to correct the UVB-1 observations for the degradation of its absolute spectral response and for sudden changes in the behaviour of the instrument. Thus, the datasets from the UVB-1



and the NILU-UV radiometers are not completely independent since the Brewer instrument was used for the calibration of both instruments.

In addition, at Thessaloniki, a CE318-N Sun Sky photometer, also known as CIMEL, provides continuously atmospheric observations through the NASA Aerosol Robotic Network (AERONET) (Balis et al., 2010). CIMEL is providing aerosol optical depth at the UV wavelength of 340 nm, amongst other aerosol properties, which is used to investigate the effects of aerosol variability at Thessaloniki on comparisons with the satellite-derived UV products.

## 3.2 Products and Algorithms

### 3.2.1 Effective UV doses from the Brewer spectrophotometer

The B086 spectra were processed by the SHICrivm algorithm and extended to 400 nm (Slaper et al., 1995). The extended spectra were validated with a collocated EKO UV-A instrument (Zempila et al., 2016a) and weighted with the action spectra for: i) the erythemal dose (McKinlay and Diffey, 1987), ii) the formation of vitamin D in the human skin (Holick et al., 2005), and iii) DNA-damage (Setlow, 1974). The corresponding effective doses have been calculated by integrating the weighted spectra over the nominal wavelength range. The 1-sigma uncertainty of the derived effective doses for the erythema and the vitamin D is estimated to be 5% since the contribution of photons with wavelengths shorter than 305 nm (where the signal may be very low) is small. However, the uncertainty in the calculated effective dose for the DNA-damage is larger at SZA greater than 60° because of the important contribution of shorter wavelengths (very low signal levels) and may reach 20% for SZAs near 80° in overcast conditions.

### 3.2.2 Effective UV doses from NILU-UV irradiances using a neural network model

A feed-forward function-approximating NN model (Hornik et al., 1989) was coded using MATLAB's object-oriented scripting language in conjunction with its Neural Network Toolbox (Beale et al., 2012). As inputs, the NN has time series vectors of NILU103 irradiance measurements at 302, 312, 320, 340 and 380 nm together with temporal variables: the SZA, the day of the week (DOW), the day of the year (DOY) and its sinusoidal components $\sin(DOY \times 2\pi/T)$ and $\cos(DOY \times 2\pi/T)$ where $T$ is the number of days in the year. As outputs, the NN calculates time series for the biological UV products resulting from B086 response weighted spectra: i.e. erythemal CIE, vitamin D and DNA-damage effective doses. The rationale behind including temporal variables in the inputs is that geophysical variables very often exhibit periodicity associated with an annual or diurnal cycle and are now commonly incorporated into atmospheric chemistry models (Kolehmainen et al., 2001). From the NILU103 data, a matrix of $n = 47,908$ co-located input-output vectors was extracted to train and validate the model. All output variables were found to correlate strongly and positively on all 5 of the irradiances ($0.922 \leq r \leq 0.995$),strongly anti-correlate with SZA ($-0.891 \leq r \leq -0.909$), and weakly anti-correlate with the temporal variables. Figure 2 shows the z-scores of the input variables and the erythemal UV dose ("CIE ") together with the pairwise linear Pearson correlation coefficient.





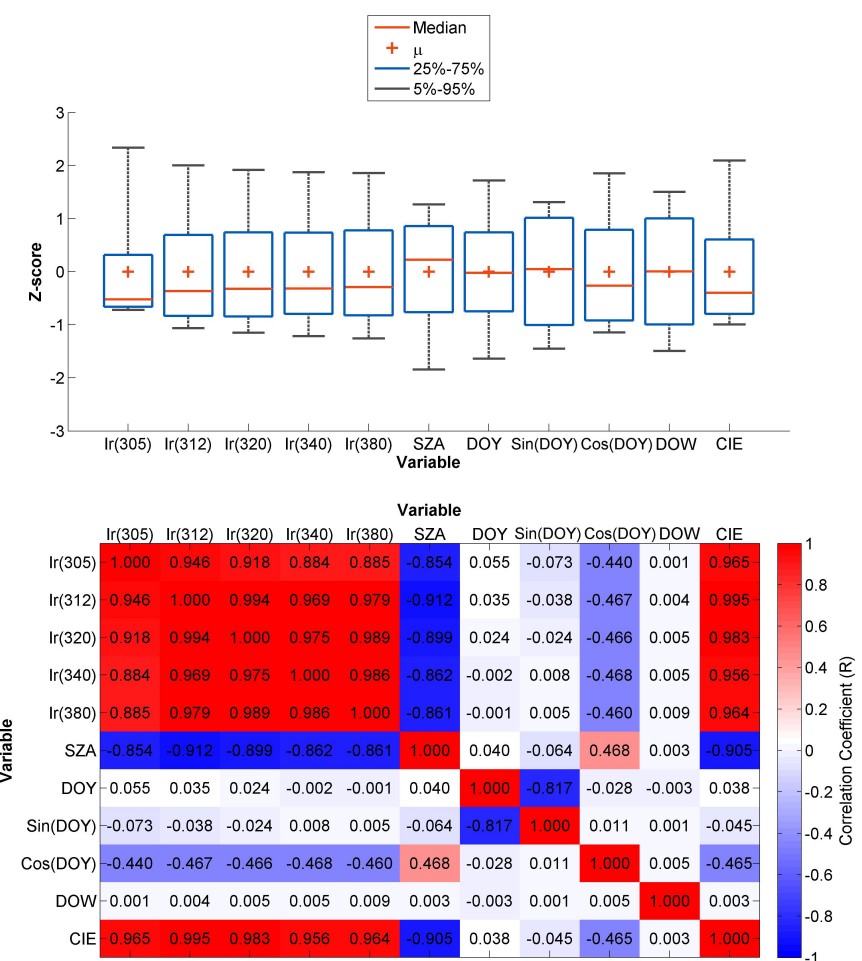

**Figure 2.** Model selection. (Top) The z-scores of the input variables and the erythemal UV dose (CIE). (Bottom) the pairwise linear Pearson correlation coefficient for each combination of the input variables and/or the output variable. The results are unnoticeably different in the case of Vitamin D and DNA damage doses. To save space, we have used abbreviated labeling of the sinusoidal terms so that $sin(DOY)$ refers explicitly to $sin(DOY \times 2\pi/T)$ etc.

The input and output vectors used in our study were connected via 2 network layers, the first containing hidden neurons with hyperbolic tangent ($tanh$) activation functions and the second containing linear activation functions. The mathematical details of this input-output structure is described in Appendix A. Key to the success of the modeling approach is signal to noise separation. The NN model is constructed using denoised time series of the NILU-UV irradiances and denoised time series of the photobiological products. Once constructed, the original (noisy) data is input to the model to calculate the photobiological outputs. In order to achieve this, we applied singular spectrum analysis to separate the signal (total trend plus periodicity) from





the total noise component for each of the irradiance and photobiological product time series (see Ghil et al. (2002) for a review of the singular spectrum analysis methodology). In this work we calculated the unbiased estimator for the lag-covariance matrix using the method of Vautard et al. (1992). The window length was rounded to $log(n)^{1.5} = 36$ following the prescription of (Kahn and Poskitt, 2010) and the minimum distance length criterion they introduce was applied. This was found to give a

consistent separation of the signal from noise for the NILU103 irradiance measurements at 302, 312, 320, 340 and 380 nm at eigenvalue ranks 9,7,7,5,5 respectively and in the case of the photobiological products, at eigenvalue ranks 7,8 and 8 respectively for CIE, vitamin D, DNA. This denoised data structure enables the NN model to determine the underlying relation between the input and output parameters most efficiently.

The optimal NN architecture was then found by minimizing the mean squared error (MSE) between the NN estimates and Brewer reference output data for each NN in a grid of 100 NN architectures where the number of hidden neurons was varied from 5 to 15 and the proportion of training data $(t/n)$ was varied from 50% to 95% in steps of 5%. The subset of t-vectors was chosen randomly with a random number generator applied to the vector of indices $[1 : n]$ and the remainder being used as a validation set that contained $(n − t)$ vectors. During each of 100 iterations of the learning process, the weights and biases of each

NN are tuned with the back-propagation optimization algorithm (Rumelhart et al., 1986) to minimize the MSE cost function over the set of input-output vectors. We have used the Bayesian regularization scheme based on a Laplace prior (Foxall et al., 2002). As a result of this initial robustness analysis, the optimal NN was found to require 13 hidden neurons and a training to validation ratio of $90\% : 10\%$ as seen in Figure 3 which also shows the result of applying the model selection procedure as well as the progression of training of the optimal NN architecture towards convergence at the horizontal asymptote for the

"best" validation MSE after 100 epochs of back-propagation learning using Bayesian regularization. Note that for the rather long time series used here, there is almost no visual dependence on the training fraction above 50% with a gradient in the optimization surface only being apparent in the direction of increasing number of neurons.

It is important to note that the optimal NN is valid for the range of parameters determined by the training data shown in

Table 1. Temporal variables other than SZA are not listed and have the following expected ranges: DOY=[0,366], $\sin(DOY \times 2\pi/T) = [−1,1]$, $\cos(DOY \times 2\pi/T) = [−1,1]$ and DOW=[1,7].

For validation, this optimally-trained NN was then fed with the remaining ("unseen") input vectors from the 10% of the training data and its estimates are compared against the target measurements of the output vector to evaluate the network performance.

The correlation of NILU103 NN estimates with target outputs was high ($r =0.988$ to $0.990$) and found to have a very low bias (0.000 to 0.011 absolute units) as shown in Figure 4. Neural-network-based estimates of retrieval uncertainty is still embryonic (see for example Ristovski et al. (2012)) due to the difficulty associated with propagating errors through a nonlinear function. In order to provide a ballpark estimate, we calculated the median absolute percentage error (MAPE) for the difference between the target values and the NN outputs and obtained the following estimates of the NN uncertainty: $\Delta(CIE) = 3.6\%$,





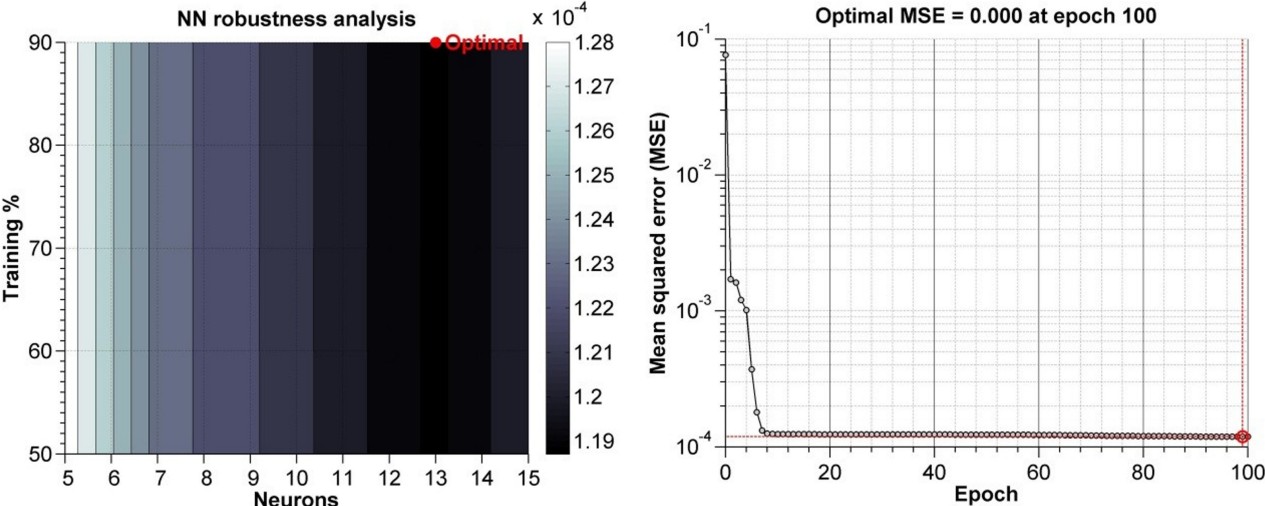

**Figure 3.** (Left) The robustness analysis on the grid of 100 NN models using the minimum validation MSE as the criterion for selection of the optimal NN architecture (which was found to have 13 hidden neurons and a training:validation data split of 90%:10%). (Right) The progress of the NN training of the optimal architecture with backpropagation iteration out to 100 iterations ("epochs") where $MSE < 1.0e^{-4}$.

**Table 1.** Range of validity of the trained optimal NN as determined by its input parameters (upper list) and output parameters (lower list).

| Parameter | Min | Max | Mean | St. Dev. |
|---|---|---|---|---|
| Ir(302) | 0 | 0.017 | 0.003 | 0.004 |
| Ir(312) | 0 | 0.229 | 0.064 | 0.055 |
| Ir(320) | 0 | 0.333 | 0.108 | 0.079 |
| Ir(340) | 0 | 0.678 | 0.252 | 0.159 |
| Ir(380) | 0 | 0.871 | 0.327 | 0.208 |
| SZA | 15.63 | 81.162 | 54.373 | 16.120 |
| CIE | 0 | 0.234 | 0.056 | 0.054 |
| vitamin D | 0 | 0.460 | 0.103 | 0.107 |
| DNA | 0 | 0.011 | 0.002 | 0.002 |

$\Delta(VitaminD) = 4.5\%$ and $\Delta(DNA) = 5.1\%$. The uncertainties seen in the NILU NN products are well aligned with the uncertainties introduced by the NILU and B086 irradiances, 5.6% and 5% respectively. An estimation of the uncertainty lying into the NILY NN products based on error propagation, results to absolute errors less than 7.5% for all three products.





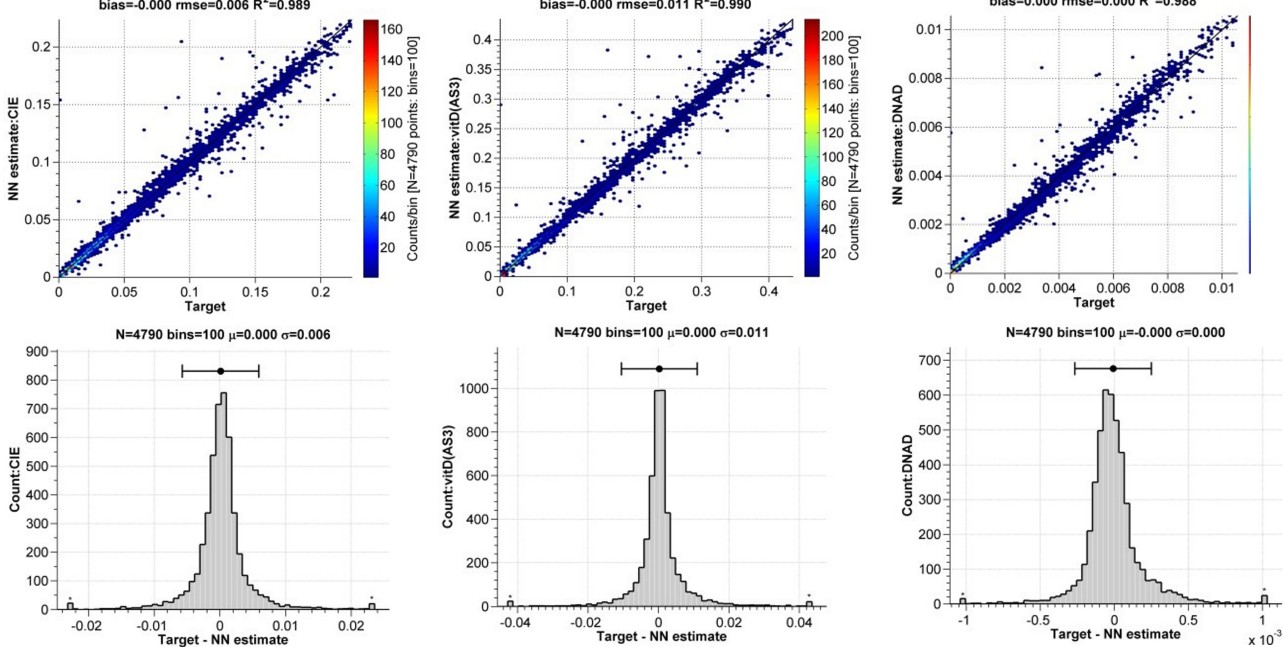

**Figure 4.** NN validation. (Upper Panels) Regression of the NILU103 NN estimates on the coincident Brewer-derived erythemal UV dose (CIE) (left), vitamin D (centre) and DNA-damage dose (right). (Lower Panels) Histograms of the difference between NN estimates and the Brewer-derived quantities. The mean ($\mu$) and standard deviation ($\sigma$) are indicated.

### 3.2.3 Effective UV doses from the UVB-1 radiometer

The Yankee UVB-1 radiometer provides measurements of the erythemal dose rates whereas the dataset's validity is monitored by coincident measurements from the double monochromator B086. Using the effective doses derived from the B086, we adopted the empirical relationship suggested by Fioletov et al. (2009) to convert erythemal irradiance to effective dose for the

5   formation of vitamin D, based on measurements of the total ozone column (TOC), and the cosine of the SZA. It was found that for UV index (UVI) values below 2, the vitamin D is overestimated significantly and should be divided by the following correction factor (cf) obtained empirically by a least squares fit to the data:

$$cf = -0.086 \cdot \text{UVI}^3 + 0.379 \cdot \text{UVI}^2 - 0.575 \cdot \text{UVI} + 1.317 \tag{5}$$

In a similar way, the DNA-damage effective doses were estimated from a more complex empirical relationship that was devel-

10   oped using data from B086 for the period $1993 - 2010$ and evaluated using data for the period $2011 - 2014$. The relationship for the DNA-damage effective doses consists of TOC, CIE, the cosine of the SZA ($cos\theta$) and the ratio between the CIE and





the climatological value of CIE on each day and SZA ($CIE_{clim}$) :

$$DNA = g(TOC, UVI, \cos\theta, UVI_{clim}) = f(CIE, TOC)/(cf1(\cos\theta) \cdot cf2(r)) \tag{6}$$

Where:

$$r = CIE/CIE_{clim} \tag{7}$$

$$f(CIE, TOC) = \frac{a_1 + s_2 \cdot CIE + a_3 \cdot CIE^2 + a_4 \cdot CIE^3 + a_5 \cdot CIE \cdot TOC + a_6 \cdot CIE^3 + a_7 \cdot TOC \cdot CIE^2}{a_8 \cdot CIE^2 + a_9 \cdot CIE + a_{10}} \tag{8}$$

$$cf1(cos\theta) = b_1 \cdot e^{b_2 \cdot cos\theta} + b_3 \cdot e^{b_4 \cdot cos\theta} \tag{9}$$

$$cf2(r) = \begin{cases} 1 & , \quad r > 2 \\ c_1 \cdot r^2 + c_2 \cdot r + c_3 & , \quad r \le 2 \end{cases} \tag{10}$$

The values of the constant terms in Eqs 8 - 10 are: $a_1 = -2.703 \times 10^{-5}$, $a_2 = 0.01245$, $a_3 = 1.428 \times 10^{-8}$, $a_4 = 0.1151$, $a_5 = -1.736 \times 10^{-5}$, $a_6 = -0.1505$, $a_7 = -9.527 \times 10^{-5}$, $a_8 = -3.523$, $a_9 = 0.9388$, $a_{10} = 0.9611$, $b_1 = 1.022$, $b_2 = -3.994$, $b_3 = 0.7306$, $b_4 = 0.2755$, $c_1 = -0.3026$, $c_2 = 0.8971$, $c_3 = 0.401$. The empirical rule given by Eq. 6 was found to be valid for UVIs greater than 0.5. The daily mean TOC from the single monochromator B005 was used in the empirical equations and in cases of missing data, daily climatological means derived from the 30-year record of B005 were used. Using the effective doses

from the double monochromator B086, we estimated that the 1-sigma uncertainty in the determination of vitamin D is smaller than 3% for UVI values greater than 2 and exceeds 10% for UVIs lower than  1. The 1-sigma uncertainty in the calculation of the effective dose for the DNA-damage is smaller than 7% for the range of used UVIs (i.e. greater than 0.5). The mean ratio between semi-simultaneous measurements of the clear sky erythemal irradiance from the B086 and the pyranometer ($\pm$ 1 minute differences between the mean time of the spectral scan and the UVB-1 measurements) for SZAs below 80° for the

period $2004 - 2014$ is $1.00 \pm 0.04$, indicating that the uncertainty in the erythemal irradiance from the pyranometer is similar to that of the Brewer B086.

### 3.3  Comparison of the NILU-UV and UVB-1 data products

Following the appropriate methodologies already discussed in Sections  3.1 and  3.2, erythemal, vitamin D and DNA-damage daily doses can be obtained from the NILU103 and an erythemal-like measuring instrument, in this case a UVB-1 radiometer.

Even though the UVB-1 data were corrected for the degradation of its absolute response with B086 data, the validity of its measurements as absolute values can be used to evaluate the performance of the NN used to derive all of the biological dose





products based on NILU-UV measurements.

In order to have comparative results for the satellite data evaluations, daily doses of all three quantities under investigation were calculated and their agreement was evaluated. For these evaluations both the UVB-1 and the NILU103 one minute data were matched in order to avoid discrepancies due to random time gaps in the original time series. Then, the daily integrals were

calculated for both NILU103 and UVB-1 datasets, without any other constrains on the data. The UVB-1 erythemal daily doses are underestimated on average by $\sim 2\%$ when compared to NILU103 retrievals, with a standard deviation of 5.39%. When limiting the data to those during which more than 70% of the original measurements were classified as cloud free, the average agreement is close to perfect (average difference of 0.48%) with a corresponding standard deviation of 4.21%. As seen in the lower panel of Figure 5(a), during the winter months UVB-1 tends to underestimate the erythemal daily doses, while during

the summer months the exact opposite behaviour is observed.

The daily integrated data for vitamin D retrievals show that there is a good agreement between the UVB-1 and NILU103 sets. In both subsets, i.e. for all- and clear-skies, respectively, the standard deviations of the differences between the two datasets are 7.43% and 5%, respectively, while the differences between the datasets are of the order of 4% for all skies and approaching zero (0.2%) for the cloud free days only. But, as observed in Figure 5(b), the number of cloud-free days is limited to only 25%

of the originally available amount of days. Again, there is a seasonal pattern for Vitamin D which is similar to the seasonal pattern observed for the daily erythemal doses.

Concerning the DNA-damage daily doses (Figure 5(c)), the comparisons show that in general UVB-1 underestimates the daily dose on average by $\sim 5\%$, with a standard deviation of about 18%. For the cloud-free days, UVB-1 show an underestimation of $\sim 2\%$ with a standard deviation of about $\sim 16\%$. The seasonal pattern observed at the lower level of Figure 5(c) is similar to the

one depicted for the aforementioned daily doses but enhanced to $\pm 20\%$, especially for the winter months where the UVB-1 significantly underestimates the doses derived from NILU103.

In Table 2 an analytical overview of the NILU103 and UVB-1 comparison statistics is presented. All three quantities present high $R^2$ values (0.99 to 1.00). The DNA data are subjected to higher sensitivity in lower wavelengths and exhibit the largest differences between NILU103 and UVB-1.

Generally, the agreement between the two instruments is quite remarkable given the different nature of the original mea-

**Table 2.** Statistical analysis of the daily integral comparisons between NILU103 and UVB-1 retrievals.

| Daily Integrals | Erythemal (%) | | Vitamin D (%) | | DNA-Damage (%) | |
|---|---|---|---|---|---|---|
| | All Skies | NILU clear | All Skies | NILU clear | All Skies | NILU clear |
| Ncounts | 3013 | 731 | 3013 | 731 | 3013 | 731 |
| $R^2$ | 1.00 | 0.99 | 1.00 | 0.99 | 0.99 | 0.99 |
| Mean (%) | -1.85 | 0.85 | -3.59 | 0.20 | -4.82 | -2.26 |
| STD (%) | 5.39 | 4.21 | 7.43 | 5.00 | 18.28 | 16.39 |

surements using different spectral resolution and different angular responses, which could be major parameters affecting the





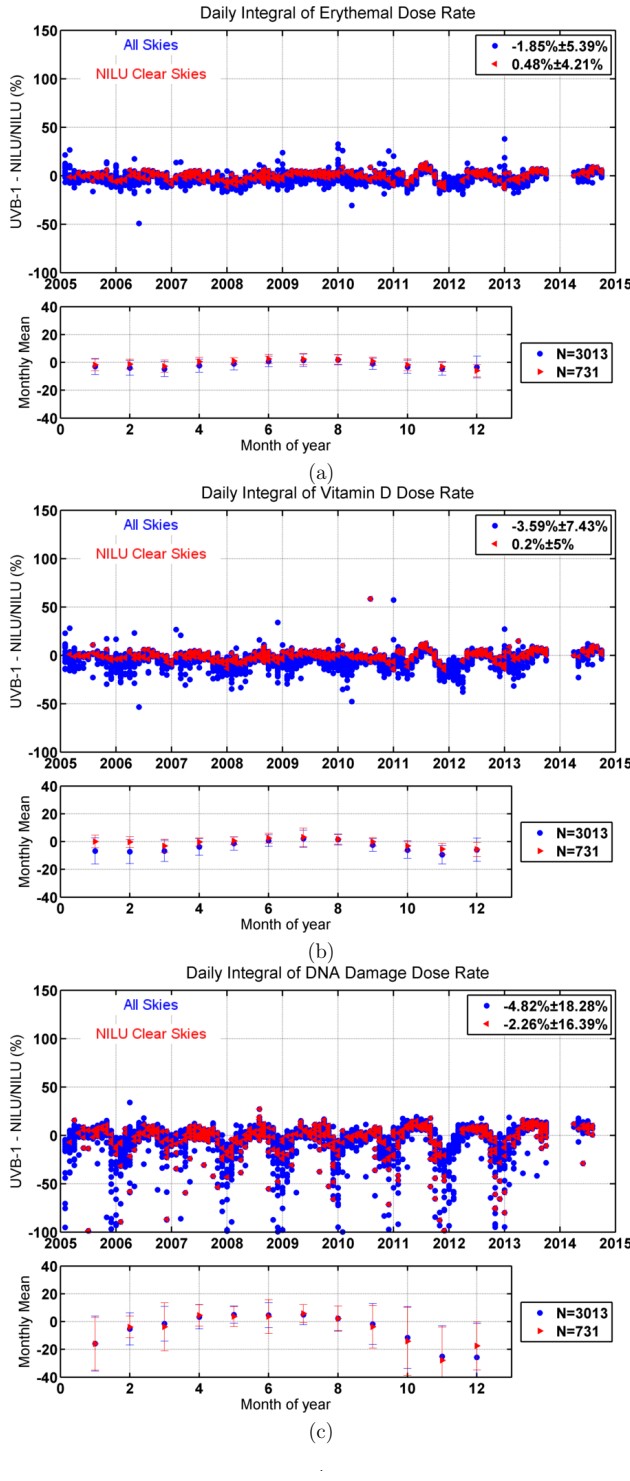

**Figure 5.** Daily integrals relative percentage differences of erythemal (a), vitamin D (b), and DNA-damage (c) doses estimates from the UVB-1 and NILU103 radiometers (upper panel) and the same datasets averaged on a monthly basis along with the 1-sigma error bars (lower panel).





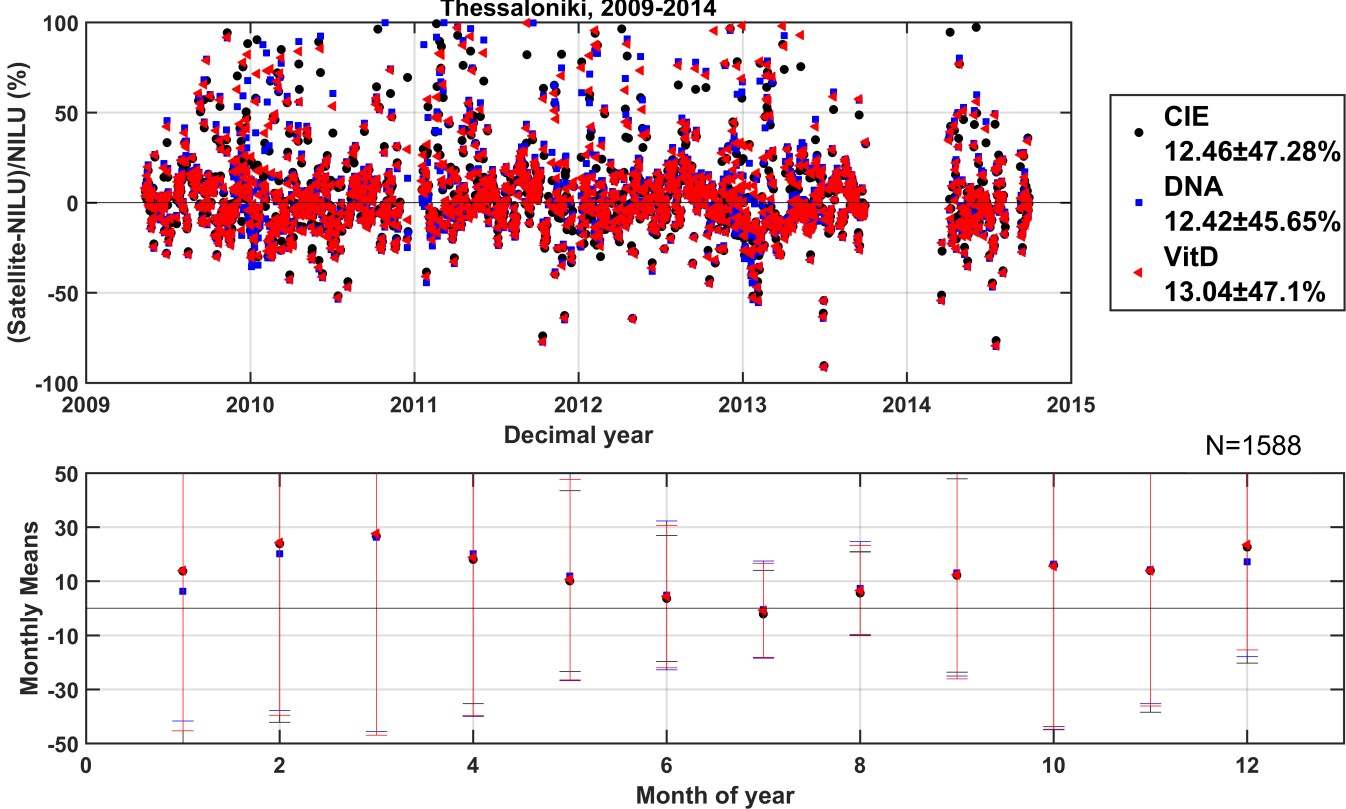

**Figure 6.** Time series of the relative percentage differences between the SCIAMACHY/GOME2A and NILU-UV effective daily doses (upper level) and the seasonality of the differences based on the average month along with the 1-sigma error bars (lower level).

comparisons, especially for the seasonal and SZA dependence, while the different retrieval methodologies could lead to further discrepancies.

## 4 Evaluation of TEMIS satellite-based UV products with NILU-UV data products

The satellite-based TEMIS UV products are evaluated for the grid cell containing Thessaloniki (grid cell centre: longitude =

5  22.75°, latitude = 40.75°). This evaluation uses a specifically reprocessed data set (version 1.4) to provide TEMIS UV dose rate values, calculated at the 10-min steps of the time integration of the daily dose UV products which are standard provided to the TEMIS data users. Time series analysis and correlation statistics are performed on the daily UV dose for erythema, vitamin D and DNA damage over a 6 year period (2009-2014). As seen in Figure 6 for all skies the TEMIS UV doses agree within 13% on average and achieve rather high correlations of 0.92, 0.93 and 0.93 for erythema, vitamin D and DNA-damage,

10  respectively. The standard deviation of the differences for the three datasets are 47.28%, 45.65% and 47.1% for erythema, vitamin D and DNA-damage, respectively. The large variations between the satellite-based and ground-based UV daily dose





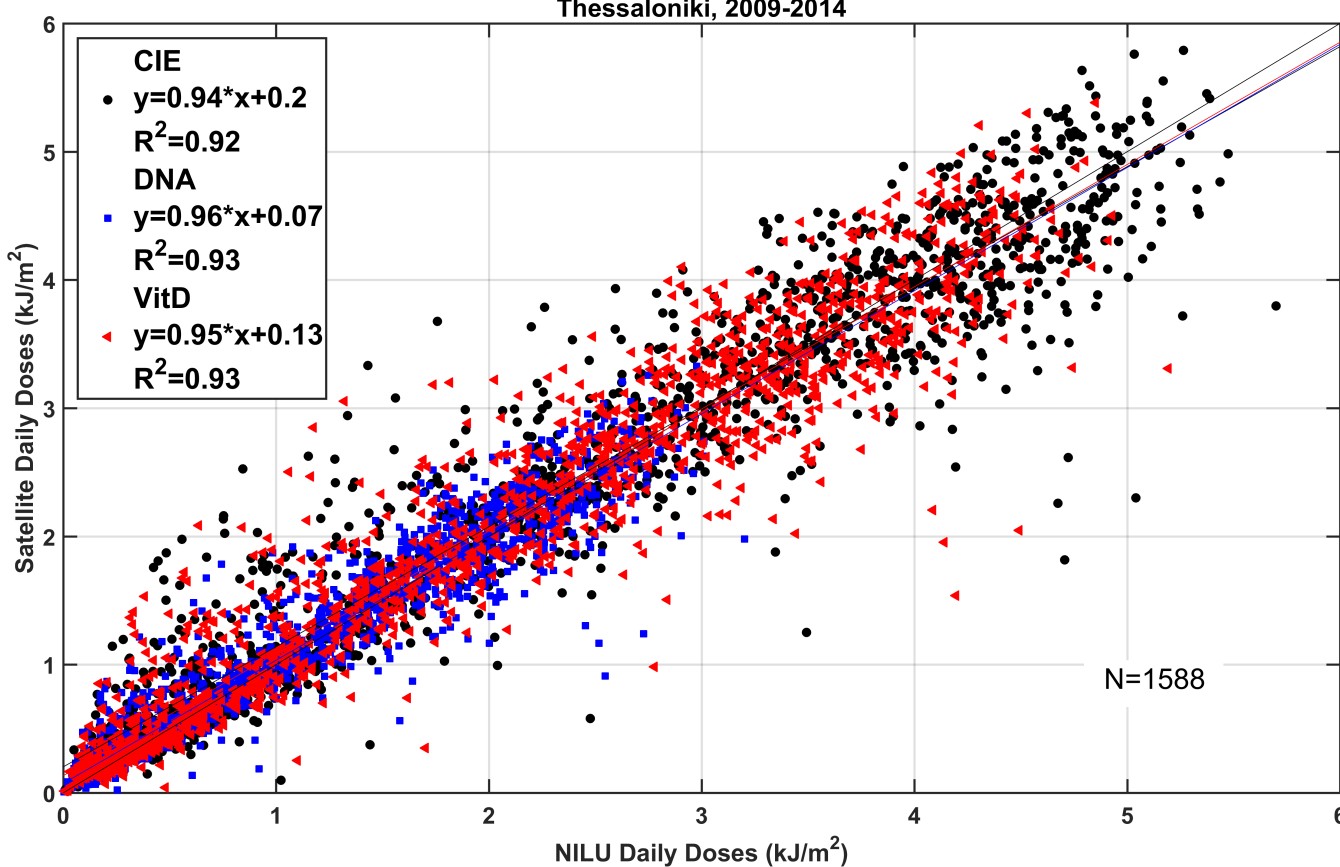

**Figure 7.** Scatter plot of daily UV dose values provided by the joint SCIA/GOME2A UV products (y-axis) and NILU103 (x-axis) in $kJ/m^2$.

data records can be attributed to different factors. For the full uncertainty budget contributions relate e.g. to the uncertainty in the B086 originally used spectra, the uncertainty caused by the application of the NILU-UV NN retrieval algorithm, the aerosol climatology assumed in the satellite-based algorithm and total ozone column retrieval errors. However, as will be demonstrated below, the greatest part of the observed spread in the ground-based and satellite-based differences in UV dose is related to the

5    representation of clouds in the satellite algorithm and selection of cloud-free days for the ground-based data sets.

     The NILU103 and TEMIS datasets have high coefficients of determination and low biases (small y-intercepts) as seen in Figure 7, while the slopes are close to unity. Although most points seem to cluster evenly around the $y = x$ line especially for the higher values, some overestimation of the satellite products at the lower values result in slopes that are slightly less than

10    unity.

   One important aspect for the evaluation is the determination of cloud-free days. The optical geometry of the two monitoring systems is different and the point measurements of the NILU at Thessaloniki compared to the $0.5° \times 0.5°$ spatial analysis of the





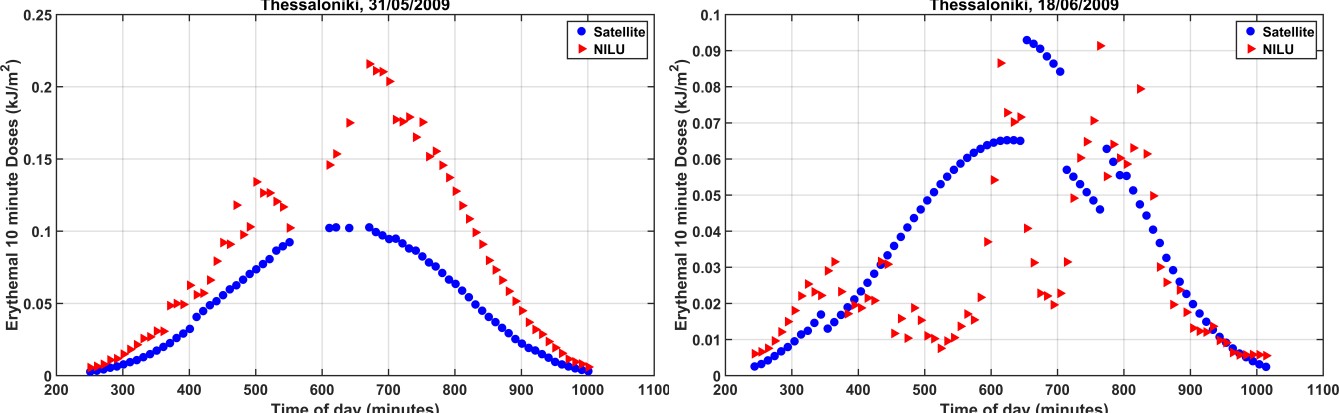

**Figure 8.** The evolution of the 10 minute erythemal dose over the day as provided by the satellite (blue circles) and at the ground (red triangles) for two days in 2009 showing a large temporal variability in cloudiness. The satellite-derived UV daily dose is lower than the NILU103-derived UV dose by 23% for the case on May, 30 2009 (left panel) while they are larger by 120% for the case on June, 18 2009 (right panel).

satellite-based product may be an important source of discrepancies. Since the satellite-based estimates are based on only one total ozone column value throughout the day, we expect that this could further increase the uncertainty in the satellite-derived daily doses estimates.

Obviously, rapidly changing cloudiness conditions can also lead to large discrepancies between the ground and satellite re-
5  trievals. Currently the TEMIS satellite doses over Europe are obtained using the cloud cover fraction per $0.5° × 0.5°$ grid cell as derived from SEVIRI/Meteosat cloud information. This information is incorporated in the TEMIS retrieval algorithm on a half-hourly basis, but the frequency of this information might need to be even higher when dealing with high frequency changing cloudiness conditions as shown in Figure 8 for two specific cloudy days at Thessaloniki. The time evolution illustrated for the two days in Figure 8 show that satellite cloud information cannot capture the rapid changes of cloudiness on these
10  days: the satellite retrievals may either overestimate or underestimate the impact of clouds. Therefore, in order to evaluate the performance of the satellite-based products, the cloudiness effects should be further analyzed. Hereto, four different cases are examined in more detail: all skies cases (whose statistical analysis is given in Figure 6); days with more than 10% of the measurements characterized as cloud free (excluding overcast days); days with more than 70% of the measurements characterized as cloud free (relatively cloudless days); and days with more than 90% of the measurements characterized as cloud free
15  (cloudless days).

An overview of the impact in limiting the percentage of cloud-free cases per day (Ncl) is provided in Figure 9 for the erythemal UV doses. The relative percentage differences clearly improve considerably when excluding the overcast days (Ncl>10%). The original 12.46% average overestimation of the satellite erythemal daily doses is reversed to 1.75% underestimation, while the standard deviation is less than 15%. When posing the 70% limitation, as applied on the (UVB-1)-NILU comparisons in





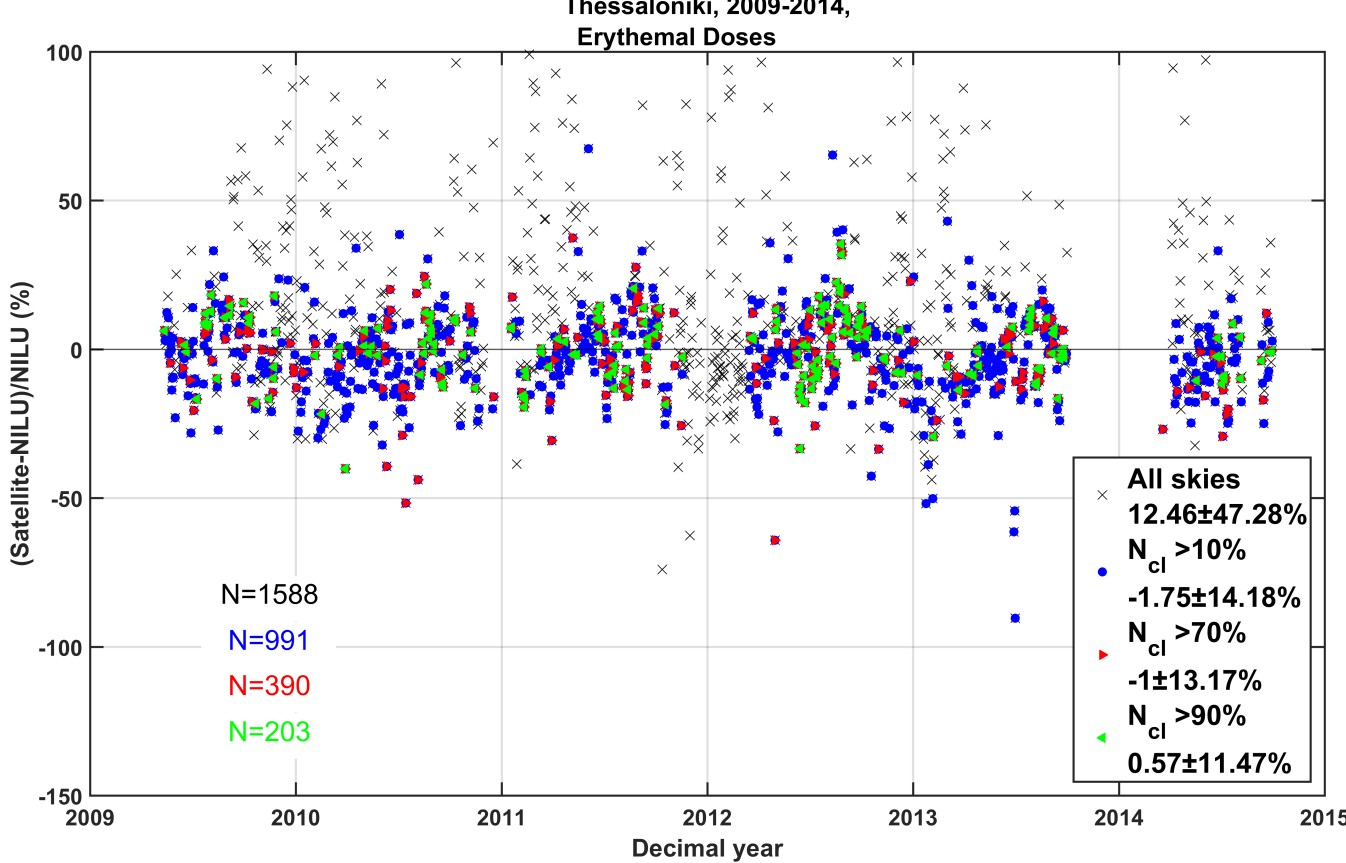

**Figure 9.** Time series of the relative differences between the satellite-based and ground-based retrieval of the UV erythemal doses; also a classification of the cloudless measurements per day is shown along with the corresponding statistics.

Section 3.3, the underestimation of the satellite erythemal doses seems to be even less while the standard deviation is similar. However, this limitation is affecting significantly the available number of days fulfilling this restriction through a reduction of number of days by 75%. On the contrary, when studying the cloudless days (Ncl>90%), the satellite product is overestimated on average by only ∼ 0.6% with a corresponding standard deviation of 11.5%. For these cloud free cases, the interpretation of

5   aerosol effects into the satellite algorithm could be an additional parameter affecting these comparisons (see bellow).

A comprehensive statistical analysis of all three UV daily doses under investigation for all cloudiness conditions is provided in Table 3. All UV doses, erythemal, vitamin D and DNA-damage, respectively, present high $R^2$ values ($\geq 0.9$) for either of the cloudiness restrictions, revealing a highly linear relationship between the two datasets. Although the satellite-based retrievals overestimate for all skies cases on average by 12.46%, 13.04% and 12.42% for erythemal, vitamin D and DNA-damage re-

10   spectively (Figure 6), the percentages are much smaller when considering only cloud-free days (in general less than 1.2%). Under mixed cloudiness conditions (Ncl>70% and >10%) satellite-based retrievals on average tend to underestimate the daily doses. As seen in table 3, the imposed cloudiness limitations do not alter much the standard deviations.



Table 3 shows that even under cloud-free days there is a scatter of almost ±13% between the two datasets for all three UV

**Table 3.** Statistical analysis of the relative percentage differences $[(Satellite - Ground)/Ground\%]$ between the satellite and ground estimates based on the cloudless instances within a day; The all skies values are given in Figures 6 and 7.

|  | Erythemal Doses | | | Vitamin D Doses | | | DNA-Damage Doses | | |
|---|---|---|---|---|---|---|---|---|---|
| Cloudless instances per day (%) | >90% | >70% | >10% | >90% | >70% | >10% | >90% | >70% | >10% |
| Ncounts | 203 | 390 | 991 | 203 | 390 | 991 | 203 | 390 | 991 |
| $R^2$ | 0.92 | 0.9 | 0.9 | 0.92 | 0.91 | 0.91 | 0.92 | 0.91 | 0.91 |
| Mean (%) | 0.57 | -1 | -1.75 | 1.22 | -0.36 | -1.40 | 1.18 | -0.34 | -1.45 |
| STD (%) | 11.47 | 13.17 | 14.18 | 12.88 | 14.48 | 15.21 | 12.18 | 13.9 | 15.76 |

doses. The seasonality seen in Figure 6 is also present when limiting the datasets to cloud-free days, implying that apart from the cloud effects, there are other factors affecting the agreement between the ground- and satellite-based UV data products. One of the causes could be variability in the aerosol load over Thessaloniki which is neglected in the satellite-based retrievals. At Thessaloniki, AOD values at 340 nm are provided by a CIMEL sun photometer for the period 2011-2014. In order to investigate the influence of aerosols on the satellite retrievals, estimations of all three UV effective doses every 10 minutes were obtained both from the satellite and NILU103 retrieval algorithms. These datasets were limited to periods where the ground-based cloud screening algorithm resulted in cloud-free cases. As seen in Figure 10 there is a strong dependence between the 10 minute doses for aerosol optical depth up to 0.4, while the differences show a slow ascending slope for aerosol loads of more than 0.4. This general pattern is in compliance with the implicit climatological AOD and SSA values applied in the satellite-based retrievals, where the AOD at 368 nm is assumed to be 0.3 and SSA is set to 0.9 (please see Section 2.2 for further details).

Model estimations performed with the model uvspec of the libRadtran library (v. 1.7) reveal that for typical aerosol optical properties for the site of Thessaloniki, differences of 0.2 between the AOD values used in the ground-based retrieval algorithm and the measured AOD, may be responsible for differences of the order of 10% between the measured and retrieved erythemal dose rates. Furthermore, other aerosol properties, like the single scattering albedo, may vary significantly over urban sites such as Thessaloniki (Bais et al., 2005) which can introduce extra uncertainties in the effect of aerosols on the estimated UV irradiances which are of the same order of magnitude as the uncertainty due to the variability in the AOD (e.g. Kazadzis et al., 2009; Fountoulakis et al., 2016a).

## 5    Discussion and Conclusions

In this work a cross-validation between ground-based measurements and evaluation of TEMIS satellite-based estimates has been performed for three important photobiological UV daily dose products: erythemal UV, vitamin D and DNA damage. The data sets to compare have been produced and compiled such to allow thoroughly discussion of their respective accuracies and limitations at the mid-latitude UV and ozone monitoring station in the Laboratory of Atmospheric Physics of the Aristotle





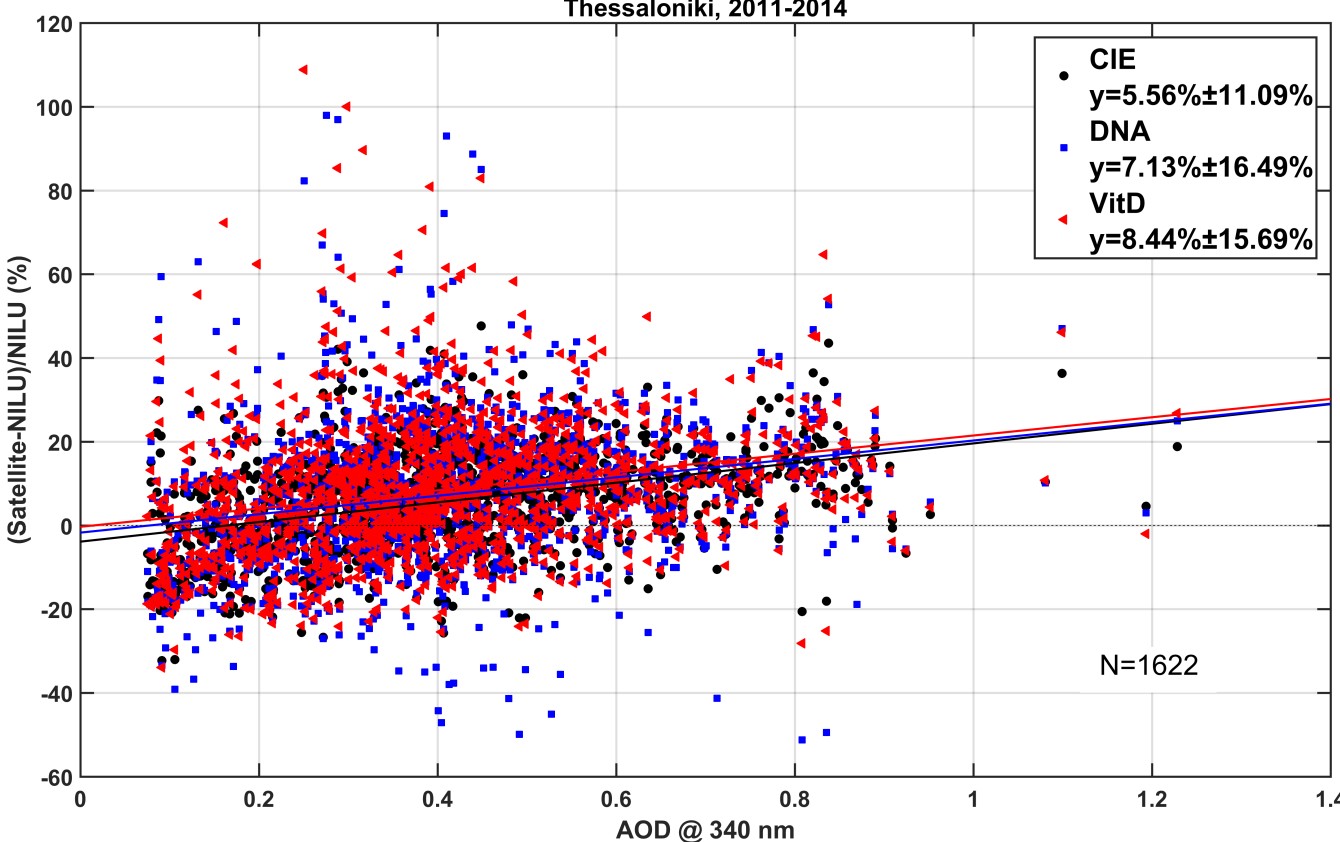

**Figure 10.** Relative differences of satellite-based and ground-based UV daily doses as a function of AOD at 340 nm for cloudless days at Thessaloniki in the period 2011-2014. The statistics are provided in the form of mean and standard deviation of the differences.

University of Thessaloniki, Greece. A neural network (NN) algorithm has been trained on NILU-UV multi-filter radiometer irradiances at 5 different wavelengths together with weighted action-spectra from a Brewer MKIII spectrophotometer to produce 1-minute time series of erythemal UV, vitamin D and DNA-damage dose rates. Further, the NN estimated erythemal UV dose rates were compared with UVB-1 calibrated UV measurements and we show how appropriate methodologies can be

5     applied to the original UVB-1 data set to also produce vitamin D and DNA-damage dose rates at the same temporal resolution as the NILU-UV instrument. In this way we could perform a ground-based verification and evaluation of the developed NN algorithm for the NILU103 measurements. The cross-validation between the NILU103 and the UVB-1 dataset revealed a very good agreement. In particular, it is found that:

- The temporally aligned NILU-UV NN and UVB-1 ground-based datasets (30,503 coincident 'all skies' dose rate data

10       records) did not show differences of more than 2% in their daily integrals and these also had a moderately low standard deviation of 5.39%.





- For vitamin D, the agreement was within 3.6% for all skies data with a standard deviation of about 7.4%, largely associated with a SZA dependence at large zenith angles. For cloud-free days this effect is reduced to about 5.0%.

- The DNA dose rates, the most demanding of the three doses discussed in this study because of their sensitivity to short wavelengths in the UV spectral region, agree to within about 5%, dropping to 2.26% for the cloud free cases.

For the evaluation of the satellite-based TEMIS UV products with the NILU-UV derived ground-based products it is found in particular that:

- The TEMIS UV daily dose products are, on average, 12.5% higher than the NILU103 daily doses under all skies. Despite the presence of a visually apparent seasonal pattern, the correlation was found to be robustly high ($R^2 = 0.92$).

- For the vitamin D (DNA-damage) UV daily doses the differences under all skies cases between the satellite- and ground-
based estimates are similar with differences of on average 13% (12.5%), again with the satellite overestimating the dose and again with very good correlation of $R^2 = 0.93$ ($R^2 = 0.93$).

It is well possible that the implicit aerosol climatology used in the satellite retrieval algorithm is at least partly contributing to higher UV doses at a moderately polluted site as Thessaloniki. Further, in the shorter wavelength part of the UV-B spectral region errors in measuring the total ozone column can have a relatively higher impact for an accurately retrieval of the DNA-
damage UV dose and Vitamin D UV dose compared to the erythemal UV dose. However, the ratios and the standard deviations for the differences in the three UV doses are similar, suggesting that the contribution of errors related to the total ozone column retrieval may not be very important. Uncertainties in the B086 spectra and the methodologies used for the calculation of the Vitamin D and DNA-damage effective doses might also be partly responsible for the observed variability, but these factors only can explain a small fraction of the total variability in the differences (in general less than 7% for all skies conditions).
Through data selections for different cloud cover conditions it was shown that the greatest part of the variability is due to the differences between the cloud cover fraction assumed in the satellite algorithm and the definition of cloud-free cases in the ground-based retrievals because the different field of view between the ground- and satellite-based instruments might lead to discrepancies regarding the cloud influences on the UV daily doses. Three clusters of cloudiness types were investigated in order to evaluate the cloud contribution on the differences between the satellite- and ground-based UV daily doses. The
introduced clusters were identified based on the percentage of cloud-free moments over a day: excluding overcast days (days with more than 10% cloudless measurements), moderate cloud-free days (days with more than 70% cloudless measurements), and cloud-free days (days more than 90% cloudless measurements).

- The number of cloud-free days limits the dataset by almost 75% and the mean relative differences are reduced for all daily UV doses. Remaining discrepancies are on average less than 1.3% for the Vitamin D and DNA-damage doses,
while the agreement for erythemal UV is on average even smaller (0.57%), revealing the notable improvement of the comparisons when excluding the cloudiness effects.

- Differences of less than 2% with moderate standard deviations ($\sim$15%) are found when excluding the overcast days, implying that the major source of the high differences observed under all skies cases can be attributed to the availability





and treatment of the cloud information, e.g. the satellite algorithm cannot distinguish between thin and thick clouds under overcast conditions

Finally, the influence of aerosol variability was investigated using the UV doses from the cloud-free days only. Coincident AOD values at 340 nm from a collocated CIMEL sunphotometer were used in order to examine the dependence of the ob-
served differences to the aerosol load at the urban site in Thessaloniki. The results showed that for AOD values up to 0.4 the contribution of aerosols to the differences in UV dose is quite significant while for even larger AOD this contribution results to slowly ascending slops. Furthermore, model estimations demonstrated that discrepancies between the measured and assumed SSA values can also lead to high differences on the retrieved irradiances which are equivalent to those attributed to the variability of AOD. Thus the discrepancies seen in the two datasets under cloud free conditions can be at least partly attributed to the
implicit aerosol information used in the satellite retrievals at the site of Thessaloniki which experiences significant variations in aerosol properties.

In conclusion, this comprehensive study has revealed the merits, limitations and accuracy of both ground-based and satellite-based estimates of erythemal UV, vitamin D and DNA-damage daily doses and underlying dose rates. Although calibration procedures, a-priori information and constraints of the methods applied in the original datasets can still limit the accuracy of
the calculated photobiological products, these types of data comparisons will remain highly important for the validation of satellite-derived UV doses and to further increase the awareness of the harmful effects of overexposure to UV radiation.

**Appendix A: Neural Network input-output theory**

The mathematical structure of the neural network model used in this work is described here.

    The NN connects a 10-parameter input vector $X = [Ir(302), Ir(312), Ir(320), Ir(340), Ir(380), SZA, DOY, \sin(DOY \times$
$2\pi/T), \cos(DOY \times 2\pi/T), DOW]^T$ through 2 layers of neurons to a 3-parameter output vector $Y = [CIE, vitaminD, DNA]^T$. Layer 1 (the ''hidden'' layer) contains $s^1$ neurons each having a nonlinear activation function $f^1 = tanh$ and Layer 2 (the ''output'' layer) contains $s^2$ neurons each having a linear activation function $f^2$. Each neuron also has a single bias $[0,1]$. $a^1$ is the vector of outputs from Layer 1 and $a^2$ is the vector of outputs from Layer 2. The vector $X$ is therefore connected to the hidden layer via a matrix of input weights $IW^{1,1}$ of size $[s^1 \times R]$ and the output of the hidden layer is connected to $s^2$ output neurons
via a matrix of layer weights $LW^{2,1}$ of size $[s^2 \times s^1]$. The vector $a^2$ for the $s^2$-outputs in vector $Y$ is the output of the NN model. The exact mathematical equation relating the outputs to the inputs is represented by the matrix equation (Taylor et al., 2014):

$$Y = f^2(LW^{2,1}f^1(IW^{1,1}X + b^1) + b^2). \tag{A1}$$

Note that the multiplication of the matrix $IW^{1,1}$ and the vector $X$ is a dot product and is equivalent to the summation over all
input connections to each neuron in the hidden layer.





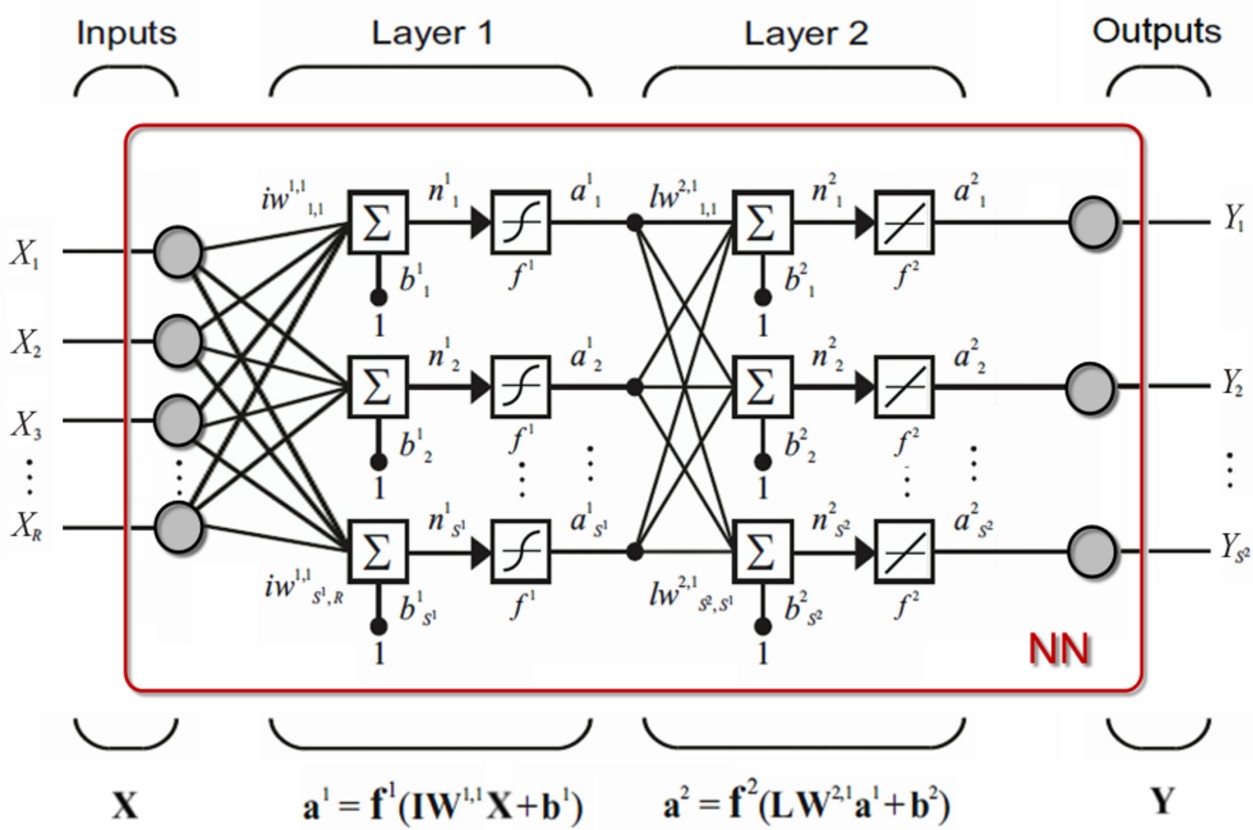

**Figure 11.** Schematic showing the neural connectivity between input and output parameters in the NILU-UV NN model.

*Acknowledgements.* The authors would like to acknowledge the National Network for the Measurement of Ultraviolet Solar Radiation, uvnet.gr.



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
