# Peer review of "TEMIS UV product validation using NILU-UV ground-based measurements in Thessaloniki, Greece"

_Atmospheric Chemistry and Physics, 2016_

## Referee Comment (RC1) · Anonymous Referee #1 · 13 Mar 2017

General comments.

The manuscript by Zempila et al. describes the validation of TEMIS UV products (specifically daily doses for erythemal UV, Vitamin D production, and DNA-damage) with ground-based measurements at Thessaloniki, Greece. Ground-based measurements are from a multi-filter radiometer, which was calibrated against a Brewer spectrophotometer using a Neural Network (NN) technique. The NN model appears to have been developed specifically for the purpose of the paper. It presents by itself a laudable addition to the suite of methods used for measuring solar radiation at the Earth's surface. The description of the NN method alone warrants publication. Comparisons of the NN model's output with Brewer measurements and measurements of a YES UVB-1

radiometer indicate that the technique works as intended, resulting in only small biases and random variations.

The topic of the paper is relevant to the audience of ACP, and the method applied are scientifically sound. I therefore recommend publication of the manuscript, provided that my general and specific comments below are taken into consideration.

Some rationale should be provided why TEMIS data were evaluated with NILU-UV measurements and not directly with Brewer measurements, which should be the most accurate. While the calibration of NILU-UV measurements against the Brewer measurement with the NN technique is a very interesting novel approach, it involves an extra step leading to an increase in the uncertainty of ground-based measurements. I realize that that NILU-UV data have much larger temporal resolution than Brewer measurements but it is not clear whether this is important considering that only daily dose data from TEMIS were evaluated. For example, are there large gaps in Brewer measurements, which would favor the NILU-UV data set? Is there an analysis that shows that the high temporal resolution of the NILU-UV data is critical for satellite data validation?

Differences between instruments are often given with a 0.01% precision. Considering that the uncertainties of all datasets are much larger, I suggest to round percentages to 0.1% throughout the paper, including the figures. This would also improve the readability of the text.

Specific comments

P2, L7: The sentence "Furthermore . . ." is confusing. It implies that the production of Vitamin D is detrimental. Mention the benefits of Vitamin D and then discuss that there may be an ideal UV exposure, which balances the harmful and beneficial effects of UV radiation!

P3, L30: I note that the 1987 CIE norm for the UV index has been updated. See: Webb,

[Figure]

Ann R., Harry Slaper, Peter Koepke, and Alois W. Schmalwieser. "Know your standard: clarifying the CIE erythema action spectrum." Photochemistry and photobiology 87, no. 2 (2011): 483-486. for details. Considering that TEMIS uses the old (1987) norm, it is OK to use this norm throughout the paper, but the new norm could be mentioned.

P4, L9: I note that the action spectrum for DNA damage suggested by Setlow (1974) is only defined for wavelengths up to 365 nm. The parameterization by Bernhard and Seckmeyer (1997), which was based on a suggestion by the NDSC steering committee (now NDACC), uses 370 as the terminal wavelength. In contrast, the spectrum drawn in Figure 1 goes up to 400 nm. The difference between the longest wavelength (365, 370, or 400 nm) is not negligible because additional contributions from the UV-A decrease the sensitivity to ozone considerably. The authors should ensure that the definition used by TEMIS is identical to that used in their work. Because the list of authors also includes colleagues that are involved in creating new versions of TEMIS products, I suggest that they carefully consider the latest definitions of the erythemal, DNA-damage, and Vitamin D action spectra when preparing a new TEMIS version.

P4, L13: Please specify the wavelength shift!

P5, L6: No. Equation (1) already defines the UV Index. So either delete this sentence or define Eq. (1) and the subsequent descriptions at erythemally weighted irradiance. In the following sentence, UVD should be calculated by integrating the erythemally weighted irradiance instead of integrating the UVI.

P5, L9: If a cloud fraction within a 0.5° x 0.5° grid cell is defined, the resolution of the satellite must be much better than 0.5° x 0.5°. What is it?

Eq. (4) is curious. If Ag is zero, fA should be 1. Yet it is 0.9775. When Ag is 1 (e.g., pristine new snow), it should be about 1.5 for erythemally weighted irradiance, yet it is only 1.3. Because Eq. (4) is part of the TEMIS code, it cannot be changed, however, it should be pointed out that the equation (which was empirically derived from measurements at two urban sites) may not be a good parameterization for large parts

of the area relevant to the TEMIS UV product, which includes Scandinavia.

P6, L25: "are less than 5.6%". Delete "less than". (The concept of "uncertainty" defines a distribution (typically normal) and 5.6% defines the width of that distribution.)

L6, L33: According to the text, only UVB-1 data were corrected for the degradation of the instrument's absolute spectral response. According to my knowledge, also NILU-UV instruments are subject to drifts. If the NILU-UV channels have drifted, as I suspect, a paragraph should be included in the manuscript describing how these drifts were corrected. How often was their calibration adjusted based on comparison with the Brewer? When comparing with the Brewer, did you take into consideration that the time associated with the Brewer measurements is different for every wavelength and did you interpolate NILU-UV measurements to the times of Brewer measurements?

Section 3.2.1: Follow-up to the previous comment: what was the time associated with a effective doses calculated from the Brewer measurements? Since a Brewer spectrum takes several minutes to record, the time is ambiguous.

Figure 2: Replace "mu" in legend with "Average"

P9, L3: What is the variable "n"?. Line 12 suggests that n is the total number of data records. However, if log(n)^1.5 = 36, n would be about 8E10 or 80 billion. This number must greatly exceed the number of NILU-UV data records!

First paragraph Section 3.2.3: The description of the calculation of effective Vitamin D dose could be improved. For example: (1) Calculate effective dose for the response function of the UVB-1 (2) Convert this instrument response function weighted dose to erythemal dose taking into account SZA and total ozone (e.g., as described on page 6, line 29). (3) Convert erythemal dose to Vitamin D dose using the parameterization suggested by Fioletov et al. (2009). (4) Apply correction (Eq. (5)).

P11, L9: The empirical relationship for DNA-damage effective dose is indeed very complex. What was the idea behind this complicated parameterization?

Eq. (6): In the second term, replace UVI with CIE.

Eq. (8): The term CIEˆ3 appears twice, with coefficient a4 and with coefficient a6. This makes little sense. CIEˆ3 should only appear one with the coefficient a4+a6= -0.0354.

P13, L10: Delete "exact"

Figure 5: The seasonal variation in 2011 appears to be much stronger than in other years. What is the reason? Wildfires? Perhaps there is something interesting that could be learned!

P15, L10 "...respectively." > "...respectively (Figure 7)."

P17, L7 and Figure 8: The right side of Figure 8 only shows 5 discontinuities. I would expect many more if cloud information is updated every half hour, as the text indicates.

P17, L13: How were cloud-free data characterized? What dataset was used to determine sky condition?

P19, L8 and Figure 10, and P22, L6: I don't see much difference in the slope for AOD < 0.4 and > 0.4. Perhaps the difference would become more obvious if the symbol size in Figure 10 were to be reduced.

Appendix A: Please specify the numbers of s1 and s2 (or the range if the numbers are not constant).

Technical corrections:

While the quality of the language is generally good, many sentences are too long and this affects the readability. Whenever possible and appropriate, the authors should reduce the length of sentences and split them in two.

P2, L5: Change "UV sunlight" to "solar radiation in the UV range". By definition, "light" should only be used to describe wavelengths visible to the human eye.

P2, L6: Delete "extreme". Mutations can technically be triggered by only one photon.
P3, L5: ". . . product services started in the 2003 and . . ."

P3, L13: "following for example changes in the operationally assimilated . . .2003) which were initially based on the . . ..and later on GOME-2 . . ."

P3, L20: ". . .SEVIRI instruments that have been operational. . ."

P3, L32: "The UVI-CIE is given as a dimensionless number . . ."

P4, L16: 'bare'? Finding a better word is indeed challenging. Perhaps: raw, uncorrected, approximate, first-guess. . .

P4, L17: ". . . is then calculated from UVI' by . . ."

P6, L10: ". . . triangular-like slit resulting in a bandwidth of 0.55 nm FWHM.

L6, L13: higher SZA > larger SZA (so not to confuse with "higher Sun")

P9, L21 and figure 3: I don't see any change in the colors of between a training fraction of 50% and 90%, consistent with the text. So if the proportion of training data has almost no effect, why is it so important to discuss this and include a figure? Is your point to illustrate that that your results are basically independent of t/n? The left figure could be simplified by plotting MSE versus the number of neurons.

P9, L33: "ballpark" > "rough" or "approximate"

P15, L10: datasets are > datasets is

P18, L7: either of the > all

P18, L11: Move "on average" to end of sentence.

P19, L24: "in the" > of

P21, L25: moments > periods

P21, L28: "limits the dataset by almost 75%" > "make up only 25% of the dataset" (if that's what you want to say)

---

## Referee Comment (RC2) · Anonymous Referee #2 · 17 Mar 2017

The study makes an important contribution to assess the accuracy in the calculation of UV doses important to health from ground-based and satellite UV data in mid-latitudes. The study focuses on the TEMIS satellite data, identifies problems in retrieving three photobiological UV dose products under cloudy conditions and points to the influence of aerosols in the retrievals using cloudless data. The study is well written, has novelty, text and figures are clear with a good flow, and the paper merits publication in ACP after few clarifications for better understanding. The points that need clarification are the sky conditions used, the seasonality, and the aerosol effect.

Specific comments

1. All-skies and clear-skies in figs 5, 6, 9: The scatter plots in figs 5 and 6 for the

case of all-skies are very different. Monthly variations and standard deviations are also very different. In fig. 5 I also note that the all-skies vs clear-skies scatter plots, and associated monthly variations and standard deviations, agree very well, something that is not seen in fig. 9. I understand that the UVB-1 and NILU data have been calibrated to a Brewer instrument before use, while the TEMIS data have not. Given that the Brewer favours measurements when the sun is not covered by clouds, can it be that this pre-calibration affects the measurements so that the all-skies in fig. 5 are not actually all-skies as in figs 6 and 9 but semi all-skies? Also, what filter do you use to define the clear-skies in fig. 5? Moreover, given that fig. 5 compares UVB-1 vs NILU data both calibrated to the same Brewer, while figs 6 and 9 compares TEMIS vs NILU data (NILU pre-calibrated to Brewer, TEMIS being not), would it make sense to calibrate also the TEMIS data to the Brewer for consistency? Potentially this pre-calibration reduces part of the variance in the original UVB-1 and NILU data, and as a consequence a better comparison is achieved between the two radiometers. I am not sure. Have you checked if the calibration to the Brewer affects the measurements denoted as all-skies? Overall I think that a clarification on the definition of all-skies and clear-skies conditions would help the reader.

2. Page 19, line 2: The seasonality of the cloud-free cases is said to match the seasonality of all-skies but it is not shown. My suggestion is to show the seasonality of the cloud-free cases because later in fig. 10 you try to explain the cause of a seasonality which is not actually shown. The seasonality can be added in fig. 9 for the lines shown in fig. 9 accordingly. I expected that the seasonality of the cloud-free cases will match the seasonality of the clear-skies shown in fig. 5 not of the all-skies shown in fig. 6. Cannot understand why since we are talking about cloud-free data. A match between the two clear-skies seasonalities would strengthen the findings about clouds affecting the TEMIS data.

3. Aerosol effect, p. 19 and fig 10: It is claimed that one of the causes for the seasonality seen in the satellite minus ground-based clear-sky differences (which is not

actually shown) is variability in the aerosol load. The authors use fig. 10 to support this. Fig. 10 shows that there is a relation between the satellite minus ground-based clear-sky differences with increasing AOD (using 10-minute time intervals), revealing a positive correlation between them, but it does not straightforwardly show the link between their seasonal variations. What is the shape of the two seasonalities and how do they match? I suggest adding an extra plot in fig. 10 (below the existing plot) showing explicitly the monthly variation of the differences vs the monthly variation of aerosols. This would strengthen the claim on p.19 line 4.

4. Page 19, lines 8-12: According to section 2.2 (p.5 lines 29-30), for AOD>0.3 the satellite UV data products will overestimate the UV index and UV dose. Indeed, the negative differences in fig. 10 tend to become positive for AOD>0.3 (indicating the satellite overestimation), but it is not clear what you mean by mentioning that the slope changes for AOD>0.4. Do you imply that there is better agreement between the satellite and ground-based data in larger AOD? I think that mentioning about two slopes confuses, unless if you clarify what you mean.

5. Is there relation between the seasonality in aerosols and the seasonality in the UVB-1 minus NILU clear-sky differences?

Minor comments:

Eq. 1: remove the unit (W/m2) from the UV index.

Page 5, lines 6-8: Is it correct that the daily UV dose is calculated from the UV index?

Page 6, line 30: It reads '... the total ozone column (TOC) and are used...'. Is it something missing from the sentence?

Page 10, line 3: correct 'NILY' to 'NILU'.

Page 15, line 9: Usually the correlation values are usually re given by the correlation coefficient R, not the R^2.

Page 18, line 5: correct 'bellow' to 'below'.

Fig. 5: Please put (a), (b) and (c) to the left side of the titles of the plots, not below the plots.

Fig 6: Indicate that the figure refers to all skies.

Fig. 7: Indicate that the figure refers to all skies. Use thicker lines for the linear lines, and use dots or dashes for the y=x line.

Fig. 10: remove the three 'y=' inside the legend since these statistics are not equations. Also, indicate that the figure refers to the >90% cloudless instances, if so.

————————————————————

---

## Author Comment (AC1) · 2 May 2017

Dear Reviewer, Please find bellow our responses to your valuable comments regarding the manuscript entitled "TEMIS UV product validation using NILU-UV ground-based measurements in Thessaloniki, Greece".

Sincerely, Dr. Melina Zempila.  

-Some rationale should be provided why TEMIS data were evaluated with NILU-UV measurements and not directly with Brewer measurements, which should be the most accurate. While the calibration of NILU-UV measurements against the Brewer measurement with the NN technique is a very interesting novel approach, it involves an

extra step leading to an increase in the uncertainty of ground-based measurements. I realize that that NILU-UV data have much larger temporal resolution than Brewer measurements but it is not clear whether this is important considering that only daily dose data from TEMIS were evaluated. For example, are there large gaps in Brewer measurements, which would favor the NILU-UV data set? Is there an analysis that shows that the high temporal resolution of the NILU-UV data is critical for satellite data validation?

We agree that Brewer data provide higher accuracy since, as the reviewer indicates, NILU data also include the uncertainty of the NN retrieval. However, for this study in order to evaluate the TEMIS daily doses we used data of 10-minute time intervals, the time resolution of the TEMIS UV dose time integration. NILU provides data with the necessary time resolution in order to acquire higher number of coincidences at the exact time of the TEMIS model estimation during a day. Unfortunately, Brewer's time frequency spans from 20 to 40 minutes (page 6 / line 23). Under cloudy conditions, this higher time resolution is considered more beneficiary for the accuracy of the comparisons. Thus, we chose to use NILU data in order to have a daily representative value. To make this clear we also added a short description on page 8, lines: 3-4. We hope that this is sufficient.

"The B086 provides measurements with a time frequency of 20 to 40 minutes, but atmospheric circumstances can change considerably within this period. It is therefore better to base the evaluation of the TEMIS UV dose rate (available at 10-minute intervals) on the NILU103 data, which have a better temporal resolution; thus they suffer much less from changes in atmospheric conditions (like clouds) during one measurement than the Brewer measurements."

- Differences between instruments are often given with a 0.01% precision. Considering that the uncertainties of all datasets are much larger, I suggest to round percentages to 0.1% throughout the paper, including the figures. This would also improve the readability of the text.

We thank you for the suggestion. We updated all pertinent graphs and text accordingly.

Specific comments - P2, L7: The sentence "Furthermore..." is confusing. It implies that the production of Vitamin D is detrimental. Mention the benefits of Vitamin D and then discuss that there may be an ideal UV exposure, which balances the harmful and beneficial effects of UV radiation!

We rephrased the sentence to "On the other hand, the cutaneous production of vitamin D, a 'vitamin' that is proven to be essential for human health, is also activated by spectral UV radiation. Hence accurate knowledge of 'safe' UV doses for humans is paramount in order to balance the harmful and beneficial effects of UV exposure."

- P3, L30: I note that the 1987 CIE norm for the UV index has been updated. See: Webb, Ann R., Harry Slaper, Peter Koepke, and Alois W. Schmalwieser. "Know your standard: clarifying the CIE erythema action spectrum." Photochemistry and photobiology 87, no.2 (2011): 483-486. for details. Considering that TEMIS uses the old (1987) norm, it is OK to use this norm throughout the paper, but the new norm could be mentioned.

We thank the reviewer for pointing us to the updated CIE spectrum. In the forthcoming upgrade of the TEMIS service, the updated CIE spectrum will be used: the expected impact on the UV index values will be small, but we consider that it is important to follow the official standard. We have rephrased the beginning of Sect. 2.2, where UVI-CIE is introduced. "In the current v1.4 TEMIS service, the UVI is based on the CIE action spectrum described by McKinlay and Diffey (1987). Webb et al. (2011) describe an improved version of that action spectrum adopted by CIE in 1998. The effect of this improvement on the UVI values is small, well below 1% except for high solar zenith angle situations (Webb et al., 2011). The improved CIE erythemal action spectrum will be included in the forthcoming upgrade (v2.0) of the TEMIS service."

- P4, L9: I note that the action spectrum for DNA damage suggested by Setlow (1974) is only defined for wavelengths up to 365 nm. The parameterization by Bernhard and

Seckmeyer (1997), which was based on a suggestion by the NDSC steering commit-
tee (now NDACC), uses 370 as the terminal wavelength. In contrast, the spectrum
drawn in Figure 1 goes up to 400 nm. The difference between the longest wavelength
(365, 370, or 400 nm) is not negligible because additional contributions from the UV-A
decrease the sensitivity to ozone considerably. The authors should ensure that the
definition used by TEMIS is identical to that used in their work. Because the list of
authors also includes colleagues that are involved in creating new versions of TEMIS
products, I suggest that they carefully consider the latest definitions of the erythemal,
DNA-damage, and Vitamin D action spectra when preparing a new TEMIS version.

We appreciate the comment. For this study we used the exact same action spectra
with the ones that TEMIS uses to avoid discrepancies due to different applied spectra
as you indicate.

- P4, L13: Please specify the wavelength shift!

We now provide this information as stated bellow: "The difference, which includes a
wavelength shift of 3 nm (the applied action spectrum peaks at 295 nm and not at 298
nm as proposed by CIE), . . ."

- P5, L6: No. Equation (1) already defines the UV Index. So either delete this sentence
or define Eq. (1) and the subsequent descriptions at erythemally weighted irradiance.

In the TEMIS processing the UVI(t) is computed in W/m2 as indicated in Eq. (1), with a
time dependent SZA(t), and as such it is used in the integration over time t to determine
the daily UVD. Only when reporting the UV index at local solar noon UVI(t=12h) the
scaling to dimensionless units is performed, which is why this sentence is present.
Describing UVI(t) in Eq.(1) as "erythemally weighted irradiance" is a good idea, thank
you – the idea has been implemented, but without "erythemally", as it is valid for all
action spectra.

- In the following sentence, UVD should be calculated by integrating the erythemally

weighted irradiance instead of integrating the UVI.

Yes, the UVD is an integration over UVI(t) over time t from sunrise to sunset, with SZA(t) dependent on time, where UVI(t) is the UV index at time t. It sounds a little confusing perhaps, but calling the UV index at local solar noon (the quantity communicated to the public) just "UV index" is actually the confusing part of this. We rephrased the whole description trying to convey this message.

- P5, L9: If a cloud fraction within a 0.5◦x 0.5◦ grid cell is defined, the resolution of the satellite must be much better than 0.5◦x 0.5◦. What is it?

The cloud fraction is derived from the MSG cloud mask. The resolution of the MSG measurements varies with latitude/longitude: along longitude 0 the resolution at latitude 30N is about 0.04 degrees, and at latitude 60N it is about 0.08 degrees.

- Eq. (4) is curious. If Ag is zero, fA should be 1. Yet it is 0.9775. When Ag is 1 (e.g., pristine new snow), it should be about 1.5 for erythemally weighted irradiance, yet it is only 1.3. Because Eq. (4) is part of the TEMIS code, it cannot be changed, however, it should be pointed out that the equation (which was empirically derived from measurements at two urban sites) may not be a good parameterisation for large parts of the area relevant to the TEMIS UV product, which includes Scandinavia.

Eq. (4) is correct because it is empirically based on the (average) ground albedo Ag of the measurement sites used for the parameterisation. This means that the albedo correction factor fA equals 1 for Ag=0.09. Many factors determine the actual enhancement of the UV due to upward diffused radiation backscattered to the surface. We do not see why this should lead to a factor equal to 1.5.

- P6, L25: "are less than 5.6%". Delete "less than". (The concept of "uncertainty" defines a distribution (typically normal) and 5.6% defines the width of that distribution.)

We deleted it.

- L6, L33: According to the text, only UVB-1 data were corrected for the degradation of

the instrument's absolute spectral response. According to my knowledge, also NILU-UV instruments are subject to drifts. If the NILU-UV channels have drifted, as I suspect, a paragraph should be included in the manuscript describing how these drifts were corrected. How often was their calibration adjusted based on comparison with the Brewer? When comparing with the Brewer, did you take into consideration that the time associated with the Brewer measurements is different for every wavelength and did you interpolate NILU-UV measurements to the times of Brewer measurements?

Thank you for the comment. On the same page, line 26 we mentioned that NILU103 measurements were calibrated with coincident B086 measured irradiances. We also added a paragraph, as suggested, to make sure that all the details are conveyed through the manuscript: "Specifically, for the calibration of NILU103 raw data, cloud free response weighted irradiances were derived from B086's measured spectra. Since B086 scans the UV solar spectrum within approximately 7 minutes, the time period needed to scan the spectral range of each NILU103's channel spectral response, is approximately 3 minutes. The coincidences of NILU103's raw data to B086's weighted spectra, were performed based on the time that B086 measured the wavelength at which each channel peaks. Subsequently, the time difference that can be introduced between the two datasets is normally less than $\pm 1$ minute. To account for this time window, the mean values of 3 consecutive NILU103 measurements were analyzed, with the central one chosen to be the closest to B086's time scan of the peak wavelength of each channel. Then, NILU103's data were corrected for possible drifts in time via a time dependent smoothing spline fit. Furthermore, the drifts of the channels were monitored through monthly lamp measurements. Both methods resulted in the same patterns for the drifted channels. After correcting for time drifts, a time independent absolute calibration factor is derived through scatter plots based on linear regression through origin. To evaluate the validity of the calibration procedures, the NILU103 calibrated data were compared once again with B086 response weighted irradiances and the timeseries were checked for time drifts and SZA dependence. By calibrating the NILU103 measurements with the B086 coincident response weighted irradiances, we

estimate that the uncertainties of the NILU103 measurements used in this study are 5.6% (Zempila et a., 2016a)."

- Section 3.2.1: Follow-up to the previous comment: what was the time associated with a effective doses calculated from the Brewer measurements? Since a Brewer spectrum takes several minutes to record, the time is ambiguous.

For the Brewer's effective doses, we considered as measuring time, the time when brewer scanned the wavelength of the peak for each action spectrum. For the DNA damage dose the starting time of the scan was taken into account. Since we used only cloud free cases, we consider that this approach doesn't introduce uncertainties larger than those of the NILU measurements themselves, even for larger SZAs. The text was updated accordingly (page 7, lines: 27-31). "The corresponding effective doses have been calculated by integrating the weighted spectra over the nominal wavelength range, while the time of measured doses was matched to the time that B086 scanned the wavelength where the highest sensitivity of each action spectrum is found. Since DNA damage action spectrum peaks at the lower measured wavelengths, the correspondent time was chosen to be the starting point of the scan. It appears that in most cases the 3 doses have time differences less than 1 minute."

- Figure 2: Replace "mu" in legend with "Average"

Thank you. We have changed the first sentence in the caption from: "Model selection. (Top) The z-scores of the input variables and the erythemal UV dose (CIE)." to: "Model selection. (Top) Boxplots of the z-scores of the input variables and the erythemal UV dose (CIE) with mean values denoted by $\mu$."

- P9, L3: What is the variable "n"? Line 12 suggests that n is the total number of data records. However, if log(n)ËĘ1.5 = 36, n would be about 8E10 or 80 billion. This number must greatly exceed the number of NILU-UV data records!

n = 47,908 is the number of co-located input-output vectors (Page 8, Line 15). To help

the reader, we have put log(n) inside brackets so that the expression now reads →
(log(n))^1.5. Precisely, this gives 35.3793 which we rounded to the next multiple of 2 to
get the value of 36 quoted in the manuscript (page 8, line: 27).

- First paragraph Section 3.2.3: The description of the calculation of effective Vitamin
D dose could be improved. For example: (1) Calculate effective dose for the response
function of the UVB-1 (2) Convert this instrument response function weighted dose to
erythemal dose taking into account SZA and total ozone (e.g., as described on page
6, line 29). (3) Convert erythemal dose to Vitamin D dose using the parameterization
suggested by Fioletov et al. (2009). (4) Apply correction (Eq. (5)).

Thank you, we enriched this section.

- P11, L9: The empirical relationship for DNA-damage effective dose is indeed very
complex. What was the idea behind this complicated parameterization?

The initial idea was to get the DNA-damage effective dose from the CIE and the main
factors that determine its levels, i.e. the TOC and the SZA in a relatively simple way
(without involving look-up tables). Although the specific quantity could be directly de-
rived from the Brewer spectra, getting it from the YES UVB-1 radiometer provides
higher temporal resolution and more accurate calculation of the daily doses. We could
not find a simpler parameterization than the one provided in the paper, for which the
calculated quantities are accurate and unbiased from the dependent variables (TOC,
SZA). Thus, although the parameterization is very complex we used it for the purposes
of the present study. We provide it in the document since it might either be directly used
by other people, or help them to find an improved, more general parameterization.

- Eq. (6): In the second term, replace UVI with CIE.

Thank you, we did.

- Eq. (8): The term CIEËĘ3 appears twice, with coefficient a4 and with coefficient a6.
This makes little sense. CIEËĘ3 should only appear one with the coefficient a4+a6=

-0.0354.

Thank you. It was a typo. The equation has been corrected properly.

- P13, L10: Delete "exact"

Thank you, we did.

- Figure 5: The seasonal variation in 2011 appears to be much stronger than in other years. What is the reason? Wildfires? Perhaps there is something interesting that could be learned!

The reviewer is correct. In early summer 2011, wildfires took place at the suburbs of the city, while data logging of the NILU's data was interrupted due to power failures resulting in less data points.

- P15, L10 "...respectively." > "...respectively (Figure 7)."

Thank you, we did.

- P17, L7 and Figure 8: The right side of Figure 8 only shows 5 discontinuities. I would expect many more if cloud information is updated every half hour, as the text indicates.

We think that discontinuities should be seen when the cloud information changes "significantly" within the 30 minutes steps. Based on the cloud information update frequency, a set of 3 points onto the graph corresponds to data with the same cloud information. If the cloud information does not change or changes slightly, discontinuities are absent of hard to be seen. Please check the changing point at around 800 (minutes) on the right hand plot of figure 8. We expect the changes to be seen within days with rapidly changing cloudiness conditions.

- P17, L13: How were cloud-free data characterized? What dataset was used to determine sky condition?

The filter we are using for defining the cloud free cases stated on page 13, line 22 is the

same for all comparisons, apart from figure 9 were we evaluate the cloud influence on the TEMIS-NILU comparisons. We added the following sentence in order to clarify this selection criterion (Page 13, lines: 23-25) "This cloud classification criterion according to which days with more than 70% abundance of cloud free measurements are characterized as cloud free, is used throughout the study, unless stated otherwise." Again on page 17, lines: 10-11, we also emphasize on this detail. "At this point it should be mentioned that for the characterization of the cloud free one-minute data, the cloud screening detector proposed by Zempila et al. (2016a) was applied on the NILU103 Photosynthetically Active Radiation (PAR) measurements"

- P19, L8 and Figure 10, and P22, L6: I don't see much difference in the slope for AOD < 0.4 and > 0.4. Perhaps the difference would become more obvious if the symbol size in Figure 10 were to be reduced.

Unfortunately resizing the marker size ends to a faint and hard to read figure. To support this statement, the linear fits of each dataset were calculated, one for AOD<=0.4 and one for AOD>0.4. For all three daily doses, CIE, DNA damage and vitamin D, the slopes are significantly larger for AOD<=0.4 than those calculated for the cases where AOD was higher than 0.4. An additional paragraph provides this information into the text (page 20, lines: 5-9). "To further testify on this aspect, linear fits were conducted for two datasets, one that comprised data with AOD$\leq$0.4 and the second with data with corresponding AOD>0.4. It was found that for all three UV effective doses, the slopes for the first imposed limitation on AOD were higher than those calculated for the second dataset. Specifically, the slopes for the two AOD limitations were found to be 44.5% and 11.7% for the CIE, 50.6% and 8.5% for the DNA damage, 46.1% and 8.3% for the vitamin D doses respectively."

Appendix A: - Please specify the numbers of s1 and s2 (or the range if the numbers are not constant).

Thank you. We have explicitly stated the values of s1 and s2 in the appropriate sentence in the Appendix (Page 23, Lines 17-18) as follows: "Layer 1 (the "hidden" layer) contains s1 = 13 neurons each having a nonlinear activation function f1 = tanh and Layer 2 (the "output" layer) contains s2 = 3 neurons each having a linear activation function f2."

Technical corrections: - While the quality of the language is generally good, many sentences are too long and this affects the readability. Whenever possible and appropriate, the authors should reduce the length of sentences and split them in two.

Thank you, we shortened the sentences where possible.

- P2, L5: Change "UV sunlight" to "solar radiation in the UV range". By definition, "light" should only be used to describe wavelengths visible to the human eye.

Thank you for the information. It was changed to solar UV radiation.

- P2, L6: Delete "extreme". Mutations can technically be triggered by only one photon.

Thank you, we did.

- P3, L5: "...product services started in the 2003 and..."

Thank you, we did change the sentence accordingly.

- P3, L13: "following for example changes in the operationally assimilated...2003) which were initially based on the....and later on GOME-2..."

Thank you, we did change the sentence accordingly.

- P3, L20: "... SEVIRI instruments that have been operational..."

Thank you, we did change the sentence accordingly.

- P3, L32: "The UVI-CIE is given as a dimensionless number..."

Thank you, we did change the "UVI-CIE" to "UVI" in order to be consistent.

- P4, L16: 'bare'? Finding a better word is indeed challenging. Perhaps: raw, uncorrected, approximate, first-guess...

We changed this to "first guess of the UV index".

- P4, L17: "... is then calculated from UVI' by..."

Thank you, we did change the sentence accordingly.

- P6, L10: "...triangular-like slit resulting in a bandwidth of 0.55 nm FWHM.

Thank you, we did change the sentence accordingly.

- L6, L13: higher SZA > larger SZA (so not to confuse with "higher Sun")

Thank you, we did change the sentence accordingly.

- P9, L21 and figure 3: I don't see any change in the colors of between a training fraction of 50% and 90%, consistent with the text. So if the proportion of training data has almost no effect, why is it so important to discuss this and include a figure? Is your point to illustrate that that your results are basically independent of t/n? The left figure could be simplified by plotting MSE versus the number of neurons.

Thank you, you are correct. As we describe in the text on Page 10, Lines 6-7, and as you note, the training MSE is not sensitive to the training fraction for large numbers of input-output vectors – rather it is sensitive to the number of neurons. While we agree that the same conclusion can be drawn by plotting MSE versus neurons, there would be a loss of information on the lack of sensitivity to training fraction. The left figure embraces both concepts in one go and is why we decided against doing this.

- P9, L33: "ballpark" > "rough" or "approximate"

Thank you, we did change that to rough.

- P15, L10: datasets are > datasets is

Thank you, we changed all occurrences.

- P18, L7: either of the > all

Thank you, we changed this point.

- P18, L11: Move "on average" to end of sentence.

Thank you, we changed this point.

- P19, L24: "in the" > of

Thank you, we changed this point.

- P21, L25: moments > periods

Thank you, we changed this point.

- P21, L28: "limits the dataset by almost 75%" > "make up only 25% of the dataset" (if that's what you want to say)

Changed to "The number of cloud-free days limits the dataset to one fourth of the original, while . . ."

Please also note the supplement to this comment:
http://www.atmos-chem-phys-discuss.net/acp-2016-1146/acp-2016-1146-AC1-supplement.pdf
* * *

---

## Author Comment (AC2) · 2 May 2017

Dear Reviewer,

Please find bellow our responses to your valuable comments regarding the manuscript entitled "TEMIS UV product validation using NILU-UV ground-based measurements in Thessaloniki, Greece".

Sincerely, Dr. Melina Zempila.

Reviewer #2: Comments and Suggestions

1. All-skies and clear-skies in figs 5, 6, 9: The scatter plots in figs 5 and 6 for the

case of all-skies are very different. Monthly variations and standard deviations are also very different. In fig. 5 I also note that the all-skies vs clear-skies scatter plots, and associated monthly variations and standard deviations, agree very well, something that is not seen in fig. 9. I understand that the UVB-1 and NILU data have been calibrated to a Brewer instrument before use, while the TEMIS data have not. Given that the Brewer favours measurements when the sun is not covered by clouds, can it be that this pre-calibration affects the measurements so that the all-skies in fig. 5 are not actually all-skies as in figs 6 and 9 but semi all-skies? Also, what filter do you use to define the clear-skies in fig. 5? Moreover, given that fig. 5 compares UVB-1 vs NILU data both calibrated to the same Brewer, while figs 6 and 9 compares TEMIS vs NILU data (NILU pre-calibrated to Brewer, TEMIS being not), would it make sense to calibrate also the TEMIS data to the Brewer for consistency? Potentially this pre-calibration reduces part of the variance in the original UVB-1 and NILU data, and as a consequence a better comparison is achieved between the two radiometers. I am not sure. Have you checked if the calibration to the Brewer affects the measurements denoted as all-skies? Overall I think that a clarification on the definition of all-skies and clear-skies conditions would help the reader.

Thank you for this comment. Here, we should notice that the UVB-1 data were not calibrated against the Brewer, but were only monitored and partially corrected for random incidences and occasional drifts caused by logging and/or electronic issues we have been experiencing during some short periods. Our intention was to prove that the NN originally applied to the NILU irradiances, results in reliable data firstly for CIE estimations, and secondly for vitamin D and DNA damage doses. We are aware that the UVB-1 data are not cosine corrected while a small overestimation of CIE takes place during the summer months. This behavior could explain the small seasonality seen in the two CIE datasets, UVB-1 and NILU (Figure 5(a)). We hope that the statement on page 13, lines: 14-16 adequately explains these aspects: "Even though the UVB-1 data were corrected for the degradation of its absolute response with B086 data, the validity of its measurements as absolute values can be used to evaluate the performance of the NN used to derive all of the biological dose products based on NILU-UV measurements."

For the NILU calibration, you are correct, we used only cloud free cases to derive the final irradiances. A detailed explanation of the NILU calibration procedures was added to the text. "Specifically, for the calibration of NILU103 raw data, cloud free response weighted irradiances were derived from B086's measured spectra. Since B086 scans the UV solar spectrum within approximately 7 minutes, the time period needed to scan the spectral range of each NILU103's channel spectral response, is approximately 3 minutes. The coincidences of NILU103's raw data to B086's weighted spectra, were performed based on the time that B086 measured the wavelength at which each channel peaks. Subsequently, the time difference that can be introduced between the two datasets is normally less than $\pm 1$ minute. To account for this time window, the mean values of 3 consecutive NILU103 measurements were analyzed, with the central one chosen to be the closest to B086's time scan of the peak wavelength of each channel. Then, NILU103's data were corrected for possible drifts in time via a time dependent smoothing spline fit. Furthermore, the drifts of the channels were monitored through monthly lamp measurements. Both methods resulted in the same patterns for the drifted channels. After correcting for time drifts, a time independent absolute calibration factor is derived through scatter plots based on linear regression through origin. To evaluate the validity of the calibration procedures, the NILU103 calibrated data were compared once again with B086 response weighted irradiances and the timeseries were checked for time drifts and SZA dependence. By calibrating the NILU103 measurements with the B086 coincident response weighted irradiances, we estimate that the uncertainties of the NILU103 measurements used in this study are 5.6% (Zempila et a., 2016a)." Based on these given details, NILU are considered to be valid for all skies cases and Brewer measurements do not affect the all skies measurements by means of implicitly excluding them. This is further testified by the fact that the agreement between UVB-1 and NILU derived CIE lies within the uncertainty of the latter, even for overcast days. Following your sequence of thoughts, we believe that now it is

more clear that UVB-1 and NILU CIE data are independent when compared in absolute values, since Brewer data served only for occasional drift correction in the UVB-1 while they were used for time drifts and absolute calibration of NILU raw data. We also agree that a pre-calibration of the TEMIS products based on Brewer measurements could take place, but the scope of this paper is to compare independent sources of estimations derived from satellite- and ground-based instruments, in our case NILU and TEMIS, in order to identify possible reasons of discrepancies between the two datasets. The comparisons performed for UVB-1 and NILU were meant to only evaluate the NN retrieval algorithm.

The filter we are using for defining the cloud free cases stated on page 13, line 22 is the same for all comparisons, apart from figure 9 were we evaluate the cloud influence on the TEMIS-NILU comparisons. We added the following sentence in order to clarify this selection criterion (Page 13, lines: 23-25) "This cloud classification criterion according to which days with more than 70% abundance of cloud free measurements are characterized as cloud free, is used throughout the study, unless stated otherwise." Again on page 17, lines: 10-11, we also emphasize on this detail. "At this point it should be mentioned that for the characterization of the cloud free one-minute data, the cloud screening detector proposed by Zempila et al. (2016a) was applied on the NILU103 Photosynthetically Active Radiation (PAR) measurements"

2. Page 19, line 2: The seasonality of the cloud-free cases is said to match the seasonality of all-skies but it is not shown. My suggestion is to show the seasonality of the cloud-free cases because later in fig. 10 you try to explain the cause of a seasonality which is not actually shown. The seasonality can be added in fig. 9 for the lines shown in fig. 9 accordingly. I expected that the seasonality of the cloud-free cases will match the seasonality of the clear-skies shown in fig. 5 not of the all-skies shown in fig. 6. Cannot understand why since we are talking about cloud-free data. A match between the two clear-skies seasonalities would strengthen the findings about clouds affecting the TEMIS data.

We thank you for the suggestion. We added the seasonality of the TEMIS/NILU comparisons for the 4 cloud classifications in the lower panel of Figure 9. Based on the findings, we cannot say that the seasonality seen in Figure 5 is the same with the one seen for the cloud free cases (Ncl>70%) in Figure 9. Although one could say that there are some similarities, when comparing these two seasonality patterns a solid conclusion is hard to be driven. We believe that these patterns are surely connected to the NILU data, but we also believe that the seasonality seen in the UVB-1/NILU comparisons is mainly due to the missing cosine correction of the UVB-1 data. On the other hand, the seasonality seen with the TEMIS/NILU comparisons can be attributed to both cosine inadequate treatment in the NILU data and/or satellite data and to the nature of the a-priori information used in the TEMIS algorithm. The pertinent paragraph was modified accordingly: "Table 3 shows that even under cloud-free days there is a scatter of almost ±13% between the two datasets for all three UV doses. The seasonality seen in Figure 6 is also present when limiting the datasets to cloud-free days, as seen in the lower panel of Figure 10, implying that apart from the cloud effects, there are other factors affecting the agreement between the ground- and satellite-based UV data products. One of the causes could be variability of aerosol load over Thessaloniki which is neglected in the satellite-based retrievals."

3. Aerosol effect, p. 19 and fig 10: It is claimed that one of the causes for the seasonality seen in the satellite minus ground-based clear-sky differences (which is not actually shown) is variability in the aerosol load. The authors use fig. 10 to support this. Fig. 10 shows that there is a relation between the satellite minus ground-based clear-sky differences with increasing AOD (using 10-minute time intervals), revealing a positive correlation between them, but it does not straightforwardly show the link between their seasonal variations. What is the shape of the two seasonalities and how do they match? I suggest adding an extra plot in fig. 10 (below the existing plot) showing explicitly the monthly variation of the differences vs the monthly variation of aerosols. This would strengthen the claim on p.19 line 4.

[Figure]

We again thank you for the suggestion. We added the seasonalities of all datasets shown in Figure 10 for the cloud free 10-minute doses. A description was also added into the text to further analyze the findings. "To further investigate the AOD impact on the comparisons, the monthly means were calculated for both AOD and relative differences. The pattern seen in the monthly means of the AOD values is in general agreement with the seasonality seen in the average monthly values of the relative percentage differences between the satellite- and ground-based 10-minute cloudless doses (Figure 10, lower panel), implying that there is a link between the two observed seasonalities."

4. Page 19, lines 8-12: According to section 2.2 (p.5 lines 29-30), for AOD>0.3 the satellite UV data products will overestimate the UV index and UV dose. Indeed, the negative differences in fig. 10 tend to become positive for AOD>0.3 (indicating the satellite overestimation), but it is not clear what you mean by mentioning that the slope changes for AOD>0.4. Do you imply that there is better agreement between the satellite and ground-based data in larger AOD? I think that mentioning about two slopes confuses, unless if you clarify what you mean.

To support this statement, the linear fits of each dataset were calculated, one for AOD<=0.4 and one for AOD>0.4. For all three daily doses, CIE, DNA damage and vitamin D, the slopes are significantly larger for AOD<=0.4 than those calculated for the cases where AOD was higher than 0.4. An additional paragraph provides this information into the text (page 20, lines:1-5). "To further testify on this aspect, linear fits were conducted for two datasets, one that comprised data with $AOD \leq 0.4$ and the second with data with corresponding AOD>0.4. It was found that for all three UV effective doses, the slopes for the first imposed limitation on AOD were higher than those calculated for the second dataset. Specifically, the slopes for the two AOD limitations were found to be 44.5% and 11.7% for the CIE, 50.6% and 8.5% for the DNA damage, 46.1% and 8.3% for the vitamin D doses respectively."

5. Is there relation between the seasonality in aerosols and the seasonality in the

UVB-1 minus NILU clear-sky differences?

To further investigate this aspect, we used the cloud free cases for both TEMIS/NILU and UVB- 1/NILU comparison results. As seen in the figure bellow, it seems that there isn't any strong correlation between the seasonality of AOD and (UVB1-NILU)/NILU% data.

Minor comments: - Eq. 1: remove the unit (W/m2) from the UV index.

In the TEMIS processing the UVI(t) is computed in W/m2 with a time dependent SZA. As such it is used in the integration over time t to determine the daily UVD. Only when reporting the UV index at local solar noon UVI(t=12h) the scaling to dimensionless units is performed, as mentioned in the sentence at page 5 / line 6 (old numbering). Hence, we leave the unit in Eq. (1); the sentence at p5/l6 has been adapted slightly.

- Page 5, lines 6-8: Is it correct that the daily UV dose is calculated from the UV index?

Yes, the UVD is an integration over UVI(t) over time t from sunrise to sunset, with SZA(t) dependent on time, where UVI(t) is the UV index at time t. It sounds a little confusing perhaps, but calling the UV index at local solar noon (the quantity communicated to the public) just "UV index" is actually the confusing part of this.

- Page 6, line 30: It reads '...the total ozone column (TOC) and are used...'. Is it something missing from the sentence?

Thank you, we rephrased that to "...the total ozone column (TOC). These factors are used to...".

- Page 10, line 3: correct 'NILY' to 'NILU'.

Thank you, we did.

- Page 15, line 9: Usually the correlation values are usually re given by the correlation coefficient R, not the RËĘ2.

We thank you for the comment. R values were added to tables 3 and 4, while additional comments on these values were included in the text along with the discussion regarding the R2 values.

- Page 18, line 5: correct 'bellow' to 'below'.

Thank you, we did.

- Fig. 5: Please put (a), (b) and (c) to the left side of the titles of the plots, not below the plots.

Thank you, we did.

- Fig 6: Indicate that the figure refers to all skies.

Thank you, we did.

- Fig. 7: Indicate that the figure refers to all skies. Use thicker lines for the linear lines, and use dots or dashes for the y=x line.

Thank you, we did.

- Fig. 10: remove the three 'y=' inside the legend since these statistics are not equations. Also, indicate that the figure refers to the >90% cloudless instances, if so.

Thank you, we revised the legend and changed the caption to: "Relative differences of satellite-based and ground-based UV 10-minute doses as a function of AOD at 340 nm for cloudless cases at Thessaloniki in the period 2011-2014. The statistics are provided in the form of mean and standard deviation of the differences (upper panel). Monthly mean values of AOD at 340 nm along with the mean monthly values of the relative differences presented in the upper panel under cloud free cases (lower panel)."

Please also note the supplement to this comment: http://www.atmos-chem-phys-discuss.net/acp-2016-1146/acp-2016-1146-AC2-supplement.pdf

---

## Author Response (AR2)

Dear editor,

Thank you for your valuable remarks.

Please find our responses on your comments marked in blue, while your comments are highlighted in bold for increased readability.

Best regards,

Dr. M. Zempila - On behalf of all co-authors.

**Comments to the Author**:

**- Page 14 line 6: why only the DNA comparison results show this seasonality ? Can that be related with their action spectrum and the fact that UVB is affecting more the DNA one ?**
The editor is right regarding the DNA dose comparisons between NILU-UV and UVB-1. For this type of comparisons the DNA damage action spectrum was normalized at the lower available wavelength as indicated on page 8 (lines: 4-5). Thus, we comment on page 15, line: 10 that:

"The seasonal pattern observed at the lower level of Figure 5(c) is similar to the one depicted for the aforementioned daily doses but enhanced to ~20%, especially for the winter months where the UVB-1 significantly underestimates the doses derived from NILU103, probably due to the fact that the DNA action spectrum peaks at shorter wavelengths."

We hope that this information clearly conveys this aspect.

**- figure 6 (down). I think the YY axis name have to be similar with the upper one.**
We now have the (Satellite-NILU)/NILU (%) Y-Axis label set in both YY axes in the subplots.

- **The seviri cloud products used have to be mentioned more. What exact cloud properties were used ?**
In section 2.1, page 3 line 18, we describe the cloud product used, NWC-SAF cloud mask, and provide a reference. In section 2.2, page 5 line 16, we describe the application of the NWC-SAF cloud mask in the TEMIS algorithm; we added the reference mentioned in 2.1 again.

On page 17 line8 (Section 4), where SEVIRI is mentioned too, we now refer back to section 2.2 to make clear which cloud properties were used.

- **You have to check the references as there are some references in the manuscript that are not mentioned in the reference section (e.g. Bais et al., 2005, Kazadzis et al., 2009, ..)**
References checked and updated accordingly.

**- figure 6b. It would be easier for the reader if you could separate black, blue and red symbols for each month in order to see more clear the value and the bars. Practically it can be done e.g. by ploting cie, vitD, DNA for March points to correspond to 2.9, 3 and 3.1 (on the XX axis) respectively.**
We followed your recommendation and changed the figure accordingly.

**figure 9: it is clear that winter months show an overestimation for cloudy conditions. Also, that cloudy conditions represent a skewed and not a normal distribution (Satellite-NILU).**
**So mean values cannot represent this in a correct way.**

We agree that mean values under all skies conditions do not represent properly the skewed distribution of the relative percentage differences [probably median values and percentiles should be a better choice]. On the other hand, most studies between satellite and ground-based estimations provide the mean and standard deviations as the metrics for comparisons. For sake of homogeneity, we didn't elaborate more on this aspect.

**AOD section:**

**I do not think that the statistics on the right side of figure 10 have much of a meaning as they are a subset of what it has been already shown before. Maybe there you can add the slopes mentioned in the text.**
**Also since the slopes are not obvious because of the amount of data in the figure, I would suggest to replace the 1600 points by X (e.g. 16) points that will represent the mean and the standard deviation of each of the (e.g. 100 points / bins) with their corresponding mean AOD.**
**Also in the caption of figure 10 there is no information about the lines drawn.**
We used 11 bins to pack the data based on the AOD values. We are now providing the linear fit into the legend information as well. The caption was updated accordingly.

**Absorption and AOD. Yes I agree with the fact that using aeronet SSA at 440nm would insert an uncertainty if used but it would be less than assuming a constant ssa =0.9 value for the whole dataset. As you already commenting is a mixed effect of AOD and SSA and I think it has to be mentioned that e.g. Arola et al have been dealing with similar studies using the aerosol absorption optical depth AOD * (1-SSA) as the quantity that affects the ground based to satellite comparison.**
To elaborate more on this aspect a short discussion was added on page 20, lines 10-13.

"Arola et al. (2009) used a monthly aerosol climatology for the AOD and SSA at 315 nm in order to correct the OMI UV irradiances for absorbing aerosols. SSA measurements in UV are not available for the under investigation period in Thessaloniki, thus a similar study, taking into account the parameter $\tau_{abs}$ = AOD·(1-SSA), cannot be performed."

**Figure 10b. There are no values for February ?**
Unfortunately, for the years 2011-2014 that coincident data of NILU and AOD at 340 nm are available, no matching was found for the month of February. That is mainly due to gaps in the AOD time series and shorter time gaps in the NILU data.

[revised manuscript text omitted]